

# Design, performance and wake characterization of a scaled wind turbine with closed-loop controls

Emmanouil M. Nanos[1], Carlo L. Bottasso[1], Filippo Campagnolo[1], Stefano Letizia[2], G. Valerio Iungo[2], and Mario A. Rotea[2]

[1]Wind Energy Institute, Technische Universität München, Garching bei München, D-85748, Germany
[2]Center for Wind Energy, Mechanical Engineering, University of Texas at Dallas, 800 W. Campbell Road, Richardson, TX 75080-3021, USA

**Correspondence:** C.L. Bottasso (carlo.bottasso@tum.de)

**Abstract.** This paper describes the design and characterization of a scaled wind turbine model, conceived to support wake and wind farm control experiments in a boundary layer wind tunnel. The turbine has a rotor diameter of 0.6 meters, and was designed to match the circulation distribution of a target conceptual full-scale turbine at its design tip speed ratio. In order to enable the testing of plant-level control strategies, the model is equipped with closed-loop pitch, torque and yaw control, and

is sensorized with integrated load cells, as well as with rotor azimuth and blade pitch encoders.

After describing the design of the turbine, its performance and wake characteristics are assessed by conducting experiments in two different wind tunnels, in laminar and turbulent conditions, collecting wake data with different measurement techniques. A large-eddy simulator coupled to an actuator-line model is used to develop a digital replica of the turbine and of the wind tunnel. For increased accuracy, the polars of the low-Reynolds airfoil used in the numerical model are tuned directly from

measurements obtained from the rotor in operation in the wind tunnel. Results indicate that the scaled turbine performs as expected, measurements are repeatable and consistent, and the wake appears to have a realistic behavior in line with expectations and with a similar slightly larger scaled model turbine. Furthermore, the predictions of the numerical model are well in line with experimental observations.

## 1 Introduction

Over the last decade, wind tunnel tests conducted with miniature wind turbine models have gained an increase attention from the research community (Bottasso and Campagnolo, 2020). The main focus of recent studies conducted with scaled turbines has been on wakes, including the characterization of the effects of the turbine operating conditions, of inflow profiles, and of thermal stability, and the testing of plant control strategies, as reported by —among many others— Chamorro and Porté-Agel (2009, 2010); Hu et al. (2012); Iungo et al. (2013); Bottasso et al. (2014b); Viola et al. (2014); Bastankhah and Porté-Agel

(2015); Howard et al. (2015); Yang et al. (2016); Campagnolo et al. (2016b); Bastankhah and Porté-Agel (2016, 2017c); Wang et al. (2017); Schreiber et al. (2017a); Campagnolo et al. (2020). Even though far from exhaustive, this list of references clearly illustrates the diversity of topics where scaled wind turbine models have been profitably used for wind energy research. Indeed, today scaled experiments in the known, controllable and repeatable conditions of the wind tunnel play a significant role in the



understanding of the physics, they support the development of mathematical models and the validation of simulation tools, and
enable the testing of new ideas and technologies in preparation for full-scale demonstration.

The vast majority of the literature focuses on the results of the experiments, but the wind turbine models are typically only
superficially described. There is only a handful of articles that address the methodology behind the design of scaled models
and/or provide some assessment of their characteristics. Trying to fill this gap is one of the goals of this work, which provides
a detailed description of the design and characterization of a new miniature wind turbine.

In Canet et al. (2021), the authors consider the laws that govern steady and transient gravo-aeroelastic scaling of wind turbine
rotors, resulting in probably the most comprehensive analysis of the problem of scaling to the present date. A similar analysis is
also developed in Bottasso and Campagnolo (2020), and forms the basis for a description of the design of scaled wind turbines
for wind tunnel testing. Scaling analysis forms also the theoretical backbone of the study presented in Wang et al. (2020a),
aimed at understanding the realism of the wakes generated by scaled models with respect to full-scale reality.

The aeroelastically scaled turbine of Bottasso et al. (2014b) and Campagnolo et al. (2014) is one of the first models described
in some detail in the literature. With a rotor diameter of 2 meters, this turbine is relatively large in size. Accordingly, it has
been primarily used in the large boundary layer test section of the wind tunnel at Politecnico di Milano, which features a
3.84 m (height) by 13.84 m (width) by 36 m (length) test section. The authors matched the relative placement of the lowest
natural frequencies of rotor, drivetrain and tower with respect to the rotor rotating frequency, and equipped the blades with
low-Reynold airfoils to guarantee a sufficiently high efficiency notwithstanding the small chord length. In addition, the model
is equipped with active individual pitch and torque control; a second-generation version of the model is also capable of active
yaw control. Strain gages measure loads on the blades, shaft and tower. Bottasso et al. (2014b) present applications related
to wind turbine controls, including emergency shutdown maneuvers, individual pitch control for load alleviation in waked
conditions, and the demonstration of an observer of the rotor inflow based on blade load harmonics (see Bertelé et al. (2021)
and references therein).

Most other models described in the literature are comparatively smaller in size. The development of a scaled model with
a rotor diameter of 0.58 m is presented in Schottler et al. (2016). The rotor aerodynamics are designed with a blade element
momentum (BEM) formulation, and the model is equipped with closed-loop active pitch and torque control. BEM is used also
in Lanfazame et al. (2016) to evaluate the effects caused on miniature wind turbine blades by the low chord-based Reynolds
flow conditions (Winslow et al., 2018). The authors designed, manufactured and tested two rotors, one of 0.45 m and one of
0.225 m of diameter. The performance of the two rotors measured in wind tunnel tests was compared against BEM and 3D
CFD simulations. Kelley et al. (2016) present a methodology for designing scaled wind turbine rotors for wake similarity.

Bastankhah and Porté-Agel (2017b) give a quite comprehensive description of a scaled wind turbine with a rotor diameter
of 0.15 m and fixed pitch. The model blades employ a cambered plate because of the low chord-based Reynolds, resulting in
a maximum power coefficient of 0.4 for a fairly low tip speed ratio (TSR) equal to 4, which probably limits the realism of
the wake immediately downstream of the rotor disk when compared to current full-scale designs. The wake of the model is
extensively characterized in Bastankhah and Porté-Agel (2017c), which report speed deficits, turbulence intensity, momentum
turbulent fluxes, meandering motions and loads on downstream machines.





A larger model is the G1 scaled turbine (Campagnolo et al., 2016b; Bottasso and Campagnolo, 2020), which has a 1.1 m diameter rotor, a power coefficient of 0.42 at a TSR of 7.5, and features closed-loop individual pitch, torque and yaw control. The rotor matches the circulation distribution of a conceptual full-scale reference at the design TSR, resulting in a realistic wake even relatively close to the rotor disk —except for the effects of the nacelle, which is comparatively larger than the one of the reference (Wang et al., 2020a). This turbine has been extensively used for wind farm control experiments and for the validation of wake models and CFD simulations (Campagnolo et al., 2016b; Schreiber et al., 2017b; Wang et al., 2019, 2020a; Bottasso and Campagnolo, 2020), here again exploiting the large dimensions of the wind tunnel in Milano to accommodate small clusters of wake-interacting turbines.

One of the principal design choices for a scaled wind turbine is its size. The literature shows that this choice implies crucial tradeoffs. In fact, smaller models alleviate the problem of blockage (Barlow et al., 1999), i.e. the effects on the flow —and hence also on the tested object— caused by the finite size of the test section. Smaller models can be tested in relatively small-size wind tunnels or, in larger facilities, allow for more numerous clusters of models to be simulated, for example in support of the study of multiple-wake interactions (Campagnolo et al., 2016b, 2020) or deep-array effects. A small size, however, limits the complexity of the model because of miniaturization and power density constraints (Bottasso and Campagnolo, 2020); additionally, a small size leads also to very low chord-based Reynolds numbers, which may limit the aerodynamic characteristics of the model. On the other hand, larger sizes enable advanced features —as for example closed-loop controls, aeroelastic scaling, and a more comprehensive sensorization—, and therefore more sophisticated applications. While a larger size relaxes somewhat the constraints due to Reynolds and miniaturization, on the other hand it also fundamentally limits the use of the models because of blockage.

Against this background, the aim of the present study is the design of a scaled turbine with similar characteristics to the G1, but with a smaller size. The main design requirements for this new turbine are the following:

– The turbine should be smaller than the G1 to expand the range of usable wind tunnels, to allow deeper array configurations than the three G1s in a row that can be tested in Milano, and it should be usable for complex terrain studies as the one described in Nanos et al. (2020).

– Despite its smaller size, the rotor should generate realistic wakes, even in the near-wake region (Wang et al., 2020a) to support the study of closely-spaced configurations.

– The model should feature closed-loop controls, to enable wind farm control studies, and should install sensors to measure loads.

It is another goal of this work to contribute to the literature, by providing a detailed description of the design, manufacturing and characterization of this new scaled wind turbine.

The material is organized as follows. Section 2 describes the design methodology and gives an overview of the model characteristics. Then, Sect. 3 presents the main performance characteristics of the turbine and its wake. Finally, Sect. 4 summarizes the main findings and gives an outlook towards future work.





## 2 Model description and design methodology

### 2.1 General description

Figure 1 shows the model with its principal components, while the main turbine characteristics are reported in Table 1. The
model features a 0.6 m three-bladed clockwise rotating rotor, and a hub height of 0.64 m. It is equipped with load sensors on
the shaft and at tower base. Collective pitch control is realized by an actuator and bevel gear system integrated in the hub,
while active yaw control is achieved with a standalone turning base. In the nacelle, two ball bearings support the shaft, which
carries a slip ring to serve the pitch actuator and shaft load sensors; an optical encoder placed immediately behind the slip ring
provides the rotor azimuthal position. A torque-meter is placed behind the aft shaft bearing, while the torque actuator is placed
at the very end of the drive train. More details on the various model sub-systems are given in the following sections.

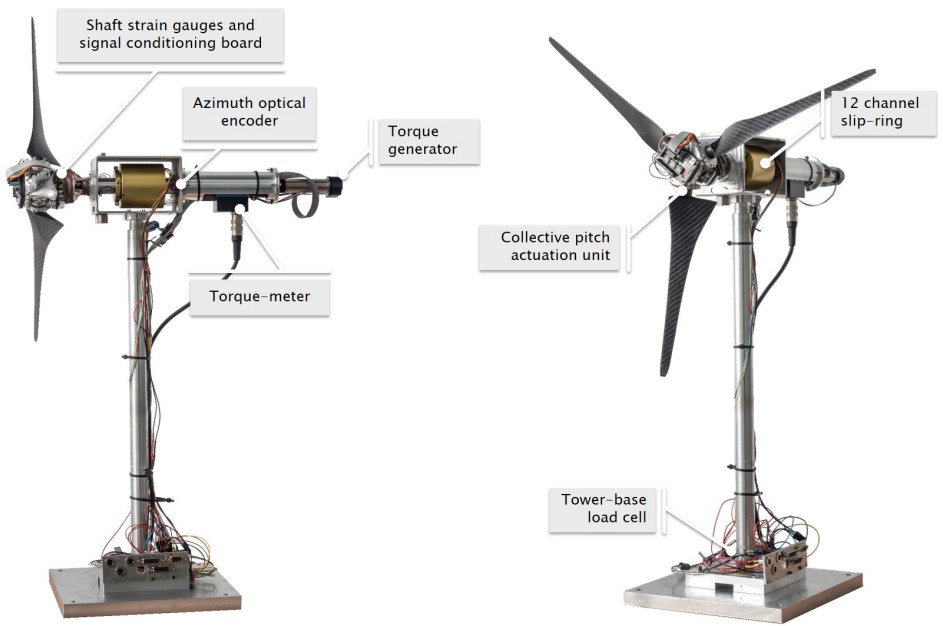

**Figure 1.** The G06 turbine with its main components.

### 2.2 Sizing of the model

As previously argued, one of the principal design choices requires the determination of the general model size, and in particular
of the rotor diameter upon which many other dimensions eventually depend. Since a compact size is a basic requirement for this
new model, the aim is to reduce the rotor diameter as much as possible. However, other design requirements impose constraints
on how small the rotor can be:



**Table 1.** Basic characteristics of the G06 scaled turbine.

| | |
|---|---|
| Nr. of blades | 3 |
| Rotation | Clockwise |
| Airfoil | RG-14 |
| Rotor diameter | 0.6 m |
| Hub height | 0.64 m |
| Rated wind speed | 10 ms$^{-1}$ |
| Rated power | 65 W |
| Active pitch control | Yes (collective) |
| Active torque control | Yes |
| Active yaw control | Yes (separate mechanism) |

– The model should be usable for simulating wake effects, including wake-induced loads, and for supporting wind farm control applications. These usage scenarios imply that:

1. Load-induced strains should be high enough to guarantee a sufficient precision of the measurements obtained from the installed transducers, notwithstanding the small aerodynamic loads. In the present case, this requirement was one of the main drivers of the geometric scaling factor.

2. The actuators and control hardware and software should be fast enough, accounting for the fact that down-scaling implies an acceleration of time with respect to the full-scale case (Bottasso et al., 2014b). This has also a strong effect on power density, which grows rapidly with time scaling (Bottasso and Campagnolo, 2020).

– Very small sizes increase the influence of manufacturing imperfections on blade aerodynamics, leading to performance deterioration and/or discrepancies among different blades (which cause rotor imbalances and differences of behavior among different models). More importantly, very small blades operate in low chord-based Reynolds conditions, which negatively influence aerodynamic performance. Wiring and miniaturization become also increasingly difficult with smaller sizes.

Other Reynolds-related conditions have an effect only for extremely small models, which however are not suitable for the present controls-oriented applications. In fact, wake behaviour is independent from the rotor-based Reynolds number when this parameter is larger than circa $10^5$ (Chamorro et al., 2012). Similarly, Reynolds-independent flows over complex terrains are obtained for terrain-height-based Reynolds numbers above $10^4$ (McAuliffe and Larose, 2012). Unless extremely small scale factors are considered, these conditions are readily met when testing in air in tunnels that produce wind speeds of the same order of magnitude of full-scale flows.

Considering these various requirements and constraints, the rotor diameter was finally chosen as D= 0.6 m.



## 2.3 Rotor aerodynamic design

### 2.3.1 General considerations

The DTU 10 MW wind turbine (Bak et al., 2013) is chosen as a baseline full-scale reference for the scaling of the G06. This machine has a rotor diameter of 178.3 m, an optimum TSR $\lambda_{\mathrm{opt}} = 8$ and a rated wind speed of 11.4 ms$^{-1}$.

The detailed aerodynamic design of the rotor aims at defining the geometry of the blade (airfoil profile(s), twist and chord distributions) that fulfills the requirements. Ideally, one would like to achieve an exact kinematic and dynamic flow similarity between scaled and reference wind turbine rotors. Kinematic similarity translates into flow streamlines that are geometrically similar, and it is directly connected to the matching of TSR. Dynamic similarity implies that the ratio of the forces acting on the model and full-scale airfoils is matched; this is a more difficult condition to achieve, as it would require matching the

chord-based Mach and Reynolds numbers (for a more in-depth discussion on the topic of scaling, see Anderson (2001) and Bottasso and Campagnolo (2020)).

For the Mach number it is sufficient to guarantee that an upper bound is not exceeded, in order to ensure the absence of compressibility effects (Bottasso and Campagnolo, 2020). The situation is, however, quite different for the Reynolds number. In fact, when testing in air, Reynolds scales as $\mathrm{Re}_M/\mathrm{Re}_F = n^2/n_t = nn_v$ (Canet et al., 2021; Bottasso and Campagnolo,

2020), where $\mathrm{Re}_M$ is the Reynolds number of the scaled model and $\mathrm{Re}_F$ the one at full scale, $n$ is the geometric scaling factor, $n_t$ is the time scaling, and $n_v = n/n_t$ is the scaling of speed. Bottasso and Campagnolo (2020) present a detailed analysis of the effects of scaling on chord-based Reynolds, including those caused by changes of chord solidity (see Fig. 1.1 of that paper). However, even a rough order-of-magnitude calculation shows the nature of the problem. In fact, scaling down the 10 MW DTU rotor to the 0.6 m diameter of the G06, implies that $n \approx 3.3 \cdot 10^{-3}$. Additionally, typical testing speeds in the

boundary layer wind tunnel in Milano are around 5 ms$^{-1}$; such a value, assuming experiments conducted around the full-scale rated wind speed, leads to $n_v \approx 1/2$. In these conditions the Reynolds mismatch is $\mathcal{O}(10^{-3})$, which is a substantial difference. Incidentally, notice that this implies $n_t = \mathcal{O}(10^{-2})$, which means that time flows about two orders of magnitude faster in the experiment than in reality. While this is a benefit in terms of data collection time (one day at full scale reduces to about 15 minutes in the tunnel), it is also a drawback in terms of real-time control, actuation rate, and sampling requirements.

Aerodynamic efficiency is defined as $E = C_L/C_D$, where $C_L$ and $C_D$ are the lift and drag coefficients, respectively. In general, typical airfoils suffer from a drastic drop in aerodynamic efficiency below a Reynolds number of about 70,000 (Selig et al., 1995) because of the formation of a laminar separation bubble. In addition, as shown in Fig. 2, at these low Reynolds a standard wind turbine airfoil as S-806 (Tangler, 1987) suffers from multiple stall-reattachment cycles even at small angles of attack. An improved behavior is obtained by ad hoc low-Reynolds airfoils, such as the RG-14 profile (Selig et al., 1995).

Notice however that the efficiency of these special airfoils is lower than the one of typical wind energy airfoils when operating at full-scale; for example, the RG-14 has an efficiency of 33.3 for a Reynolds of $5 \cdot 10^4$, while the efficiency of S-806 is about 120 for a Reynolds of $10^6$. In the end, this limits the achievable maximum power coefficient of scaled rotors. Based on these considerations, the G06 blade uses the RG-14 over its entire span, with the exception of the root region in close proximity of



the pitch bearing. Tripping, which can be employed for triggering the boundary layer transition and eliminate or reduce the
laminar bubble (Selig and McGranahan, 2004), is not used on the G06 blades because it is not effective on low-camber airfoils.

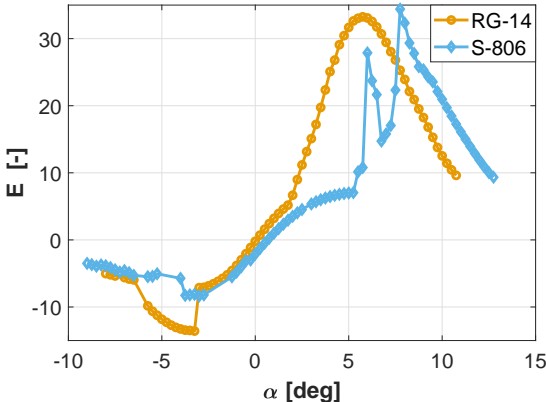

**Figure 2.** Airfoil efficiency $E$ vs. angle of attack $\alpha$ for the high Reynolds S-806 airfoil (blue line and $+$ symbols) and the low Reynolds RG-14 airfoil (red line and $\circ$ symbols). The characteristics of both airfoils are evaluated at Re$= 50,000$ using Xfoil (Drela).

Wake similarity is obtained by matching the geometry and strength of the vortex filaments released by the blades (Canet et al., 2021; Bottasso and Campagnolo, 2020).

The correct vortex geometry is obtained by ensuring kinematic similarity, i.e. matching the TSR $\lambda = \Omega R/U$, where $\Omega$ is the rotor speed, $R = D/2$ the rotor radius, and $U$ the ambient wind speed.

On the other hand, the correct strength of the vortex filaments is obtained by matching the spanwise circulation distribution. According to Prandtl lifting line theory (Anderson, 2001), a blade can be represented as a superposition of vortices of strength $\Gamma$ (circulation). Due to Helmholtz's theorem, each vortex extends as two free vortices trailing downstream all the way to infinity. Biot-Savart law states that each filament induces a velocity $w = \Gamma/4\pi h$ at an arbitrary point located at a filament-orthogonal distance $h$ away. Eventually, the velocity at any point in the flowfield is the combination of the free-stream velocity and the
velocities induced by all vortex filaments at that point. The lift per unit span $\mathrm{d}L$ at a blade segment of span $\mathrm{d}r$ is related to the circulation $\Gamma$ of this segment by the Kutta-Joukowski theorem:

$$\mathrm{d}L = \rho W \Gamma \mathrm{d}r, \tag{1}$$

where $\rho$ is air density, $W$ is the relative flow velocity, lift is $\mathrm{d}L = 1/2\rho W^2 c C_L \mathrm{d}r$, where $c$ is the chord length. Inserting the expression for lift into Eq. (1) and nondimensionalizing by the free stream velocity and the rotor radius yields:

$$\Gamma' = \frac{C_L}{2} \frac{W}{U} \frac{c}{R}. \tag{2}$$

Wake similarity is obtained by matching the circulation distribution, as expressed by Eq. (2), along the span of the scaled and reference turbines.



### 2.3.2 Rotor design methodology

The rotor design problem is formulated as the following constrained optimization:

$$C_P^* = \max_{\theta,c} C_P \left( \theta, c, D, \lambda_{\mathrm{opt}}, \Omega_{\mathrm{scaled,rated}} \right), \tag{3a}$$

$$\text{s.t.:} \quad \Gamma_i' = \Gamma_i'^{\mathrm{ref}}, \quad i = [1, N], \tag{3b}$$

$$\mathrm{Re}_{av.} \geq 70{,}000, \tag{3c}$$

where the power coefficient is $C_P = P/(0.5\rho U^3 \pi R^2)$, $P$ indicates power, and subscript $i$ stands for a generic spanwise control section along the blade.

The optimization problem seeks the blade twist $\theta$ and chord $c$ distributions that maximize the rotor power coefficient $C_P$. The power coefficient is estimated by using BEM (Burton et al., 2001), and chord and twist distributions are discretized using splines. The optimal design problem is solved using the Interior Point Method, as implemented in Matlab (Mathworks, 2019).

The optimization is constrained by the matching of the nondimensional circulation at a number $N$ of spanwise control stations. A second constraint condition sets a lower limit for the average Reynolds number along the blade, which can be met by the optimizer by locally increasing the chord with respect to the one of the reference turbine. Since there is no explicit constraint on solidity, it should be noted that the maximum power coefficient of the scaled rotor is not necessarily coincident with the optimum TSR $\lambda_{\mathrm{opt}}$ of the reference rotor (Bottasso and Campagnolo, 2020), which is however not a concern in this case.

The rated rotor speed of the scaled model, $\Omega_{\mathrm{scaled,rated}} = 2{,}250$ rpm, was primarily determined by the requirement to avoid compressible effects over the blade, as expressed by the condition $\Omega_{\mathrm{scaled,rated}} R/c_s \leq 0.3$, $c_s$ being the speed of sound.

### 2.3.3 Blade shape and fabrication

The methodology described in the previous section resulted in the blade geometry shown in Fig. 3 in terms of chord and twist distributions.

Criteria for the choice of the blade material and of the manufacturing technology were rigidity (to avoid deformations in operation), high precision and consistency (to ensure similar blades), and lifetime (on account of the high rotor speed and hence large expected number of cycles).

The blade comprises of three parts: the carbon fiber skin, which determines the external shape of the blade and carries the loads, a foam filler in Rohacell, and an aluminum root used to connect with the pinion gear.

The manufacturing process uses a high-precision aluminum female mold in two halves. Each mold half is laminated with carbon fiber sheets of 0.25 mm of thickness, using two plies close to the root and one from mid-span onwards towards the tip. The metal root is then inserted into position. The Rohacell foam filler is placed on the molds, which are then joined together and placed in the oven for the curing process. The Rohacell foam expands during curing, pushing the carbon fiber sheets onto the molds, thereby ensuring a smooth external surface.





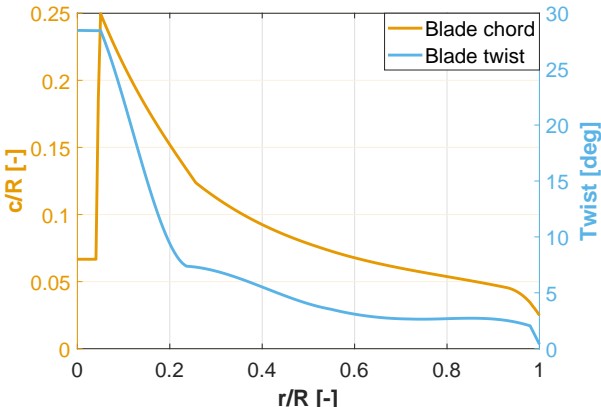

**Figure 3.** Chord and twist distributions along the blade.

## 2.4 Actuators

### 2.4.1 Pitch actuation mechanism

Given the relatively small size of the G06, an individual pitch control system would increase cost and complexity. Considering also its typical use cases, a collective pitch control system was chosen for this model.

The pitch mechanism is realized through a bevel gear system, featuring a crown and three pinions (see Fig. 4). The crown is connected through a flexible coupling with a Maxon gearhead, and each pinion is connected with its own respective blade. The gearhead has a 84:1 ratio, and it is driven by a Maxon 30 W DC motor. According to the manufacturer, a $1.3°$ backlash is to be expected for the gearhead. Given that the bevel gear ratio is 27:15, this gearhead backlash translates into a $2°$ play at the blade pitch angle, which is unacceptable. To eliminate this backlash, each blade is attached to a torsional spring. The spring constant and its position ensure that the spring is always under tension within the pitch angle operational range, and that the applied torque is always higher than the aerodynamic pitching moment on the blade. Consequently, the loading direction on the gearhead is always the same, resulting in a solution that presents no backlash of the blade pitch motion.

The pitch motor is controlled through a two-channel encoder, thus only relative angular displacements are possible. The absolute pitch rotation of the blade is obtained by Hall sensors, as described later in Sect. 2.5.2.

To verify the suitability of the actuator, the pitch actuation system dynamics were modeled in Simulink. The maximum continuous pitch rate is $550°\,\mathrm{s}^{-1}$. Considering that the time scale factor between the G06 and the full-scale reference is $n_t \approx 1/240$, this corresponds to a full-scale pitch rate of approximately $2.3°\,\mathrm{s}^{-1}$. This value is smaller than the typical maximum operational pitch rate of full-scale turbines, which is approximately in the range $[6-9]°\,\mathrm{s}^{-1}$. Aeroelastic simulations of the DTU 10 MW turbine were conducted in the full-load regime (region III) with a turbulence intensity of $10\%$. The analysis of these simulations indicates that the pitch actuation exceeds $2.3°\,\mathrm{s}^{-1}$ for only $5\%$ of the time. Based on these results, the speed of the pitch actuator was deemed acceptable.



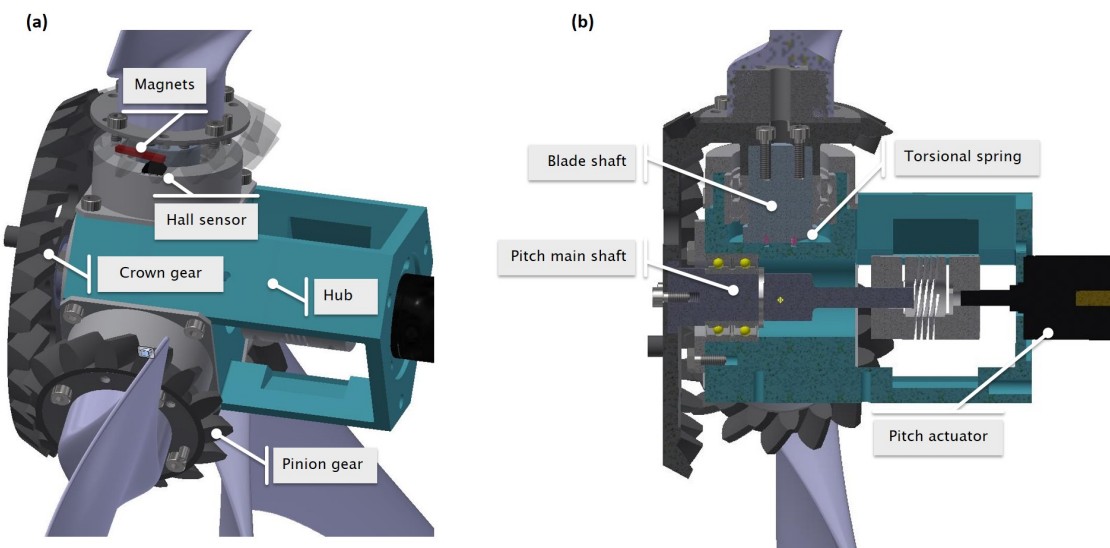

**Figure 4.** Different views of the hub assembly. View with one transparent blade gear, to show the magnets and Hall sensor **(a)**. View where the assembly has been cut to reveal hidden elements **(b)**.

With this pitch rate, the G06 actuation system is suitable also for other non-standard applications, such as dynamic induction wind farm control (Frederik et al., 2020; Munters and Meyers, 2018). For example, a high (above the optimal) Strouhal frequency $St = fD/U = 0.6$ and a pitch amplitude of $6°$ are well within the limits of the present system.

### 2.4.2   Torque actuator

The torque actuator provides either a torque or a speed operation mode, depending on the application. In torque mode, the actuator plays the same role of the generator in a real wind turbine, whereas in speed mode it provides the torque that is necessary to spin the rotor at a desired angular velocity. The actuator is a Maxon DC 120 W motor, equipped with a gearhead with a 4.4:1 gear ratio, produced by the same manufacturer. The motor is controlled through an analog Maxon ESCON Module 50/5 controller, which allows for the user to select between the two modes (torque or speed) of operation.

When the motor works as a generator, current flows from the motor to the controller and from there to the power supply. To dissipate this flow of current, the motor controller is connected in parallel with an 8 Ohm resistor capable of dissipating up to 100 W of power.

### 2.4.3   Yaw actuation system

Due to the small size of the G06 model, integrating the yaw mechanism into the tower —as done for the G1 and G2 turbines— would increase the tower diameter. An excessively out-of-scale tower creates a wider wake and has a mismatched vortex





shedding (Wang et al., 2020a), in turn affecting the turbine wake. To avoid this problem, the yaw actuation mechanism is realized through a separate turning base on which the G06 is mounted.

This solution not only enables the design of a thinner tower, but also decouples the yaw mechanism from the turbine itself, making the assembly process easier and faster. Despite the physical decoupling, the yaw actuation mechanism is controlled through the same control hardware and software as the other models of the TUM family of scaled wind turbines.

**2.5   Sensorization of the model**

**2.5.1   Force and torque sensors**

The G06 is equipped with strain sensors to measure bending and torsional moments on its shaft. To this end, three full-strain gauge bridges are located immediately in front of the first bearing (Fig. 5a); two bridges are sensitive to shaft bending, whereas the third is sensitive to torsion. Bending information is used for assessing the loading on the turbine, optionally
after transforming the rotating signals into a fixed frame of reference. Torsional loads are used for the evaluation of the rotor performance by measuring the aerodynamic torque. Each bridge is connected to a conditioning board mounted on the hub. Signals and power to/from the conditioning boards are transferred to the control unit through a 12-channel slip ring. In addition to the strain gauges, a high-precision commercial torque-meter (Lorenz Messtechnik GmbH) is placed between the aft bearing and the generator. The torque-meter has a higher precision and sampling frequency than the strain gauges, but its readings are
affected by the friction in the bearings and the slip ring. This friction, which depends on various factors and may change over time because of temperature and wear, can be estimated by the difference between the readings of the strain gauges and the torque-meter.

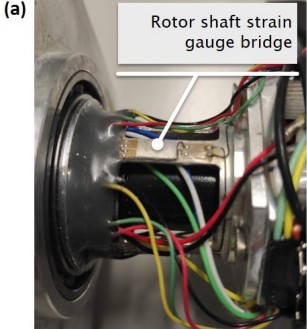
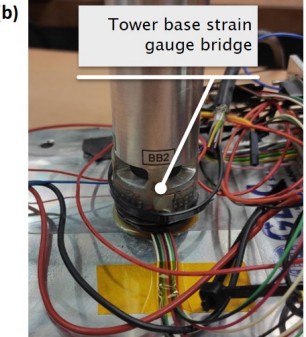
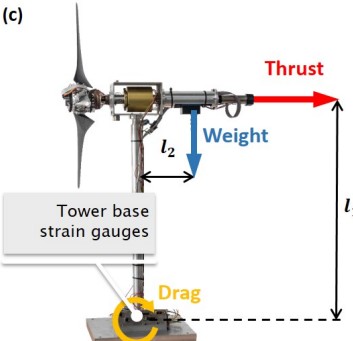

**Figure 5.** The rotor shaft, with its strain gauge bridge **(a)**; tower base, with its own integrated load cell **(b)**; schematic representation of the forces acting in the fore-aft direction on the model, and the respective moments induced at the strain gauge position **(c)**.

Two additional full bridges are placed at the base of the tower to measure fore-aft and side-side bending (Fig. 5b). The thrust generated by the rotor can be estimated from the former bending moment. In fact, as shown in Fig. 5c, the total fore-aft moment
$M_o$ measured by the strain gauges is the sum of the moments due to the rotor thrust $M_T$, the tower and nacelle drag $M_D$, and



the nacelle weight $M_G$, i.e.

$$M_o = M_T + M_D + M_G, \tag{4}$$

where $M_T = T\,l_1$, $l_1$ being the moment arm of thrust $T$, which is assumed to be applied at the rotor center. The values of $M_D$ and $M_G$ are determined off-line with dedicated measurements. For calculating $M_D$, the blades are removed and the model is

placed in thewind tunnel, where measurements at various wind speeds are taken. For calculating $M_G$, a single measurement without wind is sufficient.

The shaft and tower bridges are calibrated prior to each experiment by the use of known loads, measuring the voltage and correlating loads and output via a linear regression.

### 2.5.2 Position sensors

Two kinds of position sensors are used in the model: Hall sensors and rotary optical encoders. Both the torque and pitch motors have their own internal optical encoders, which are used by the respective internal controllers.

The pitch motor is used to rotate the blades to a specific angular position, but can only be commanded through a relative angular displacement. The absolute orientation of the blades is obtained by a Hall sensor. As shown in Fig. 4a, the Hall sensor is stationary and placed on the casing of the blade bearings, while magnets are placed on the bevel gear and rotate together

with the blades. The relationship between Hall sensor output and blade pitch angle is determined by a calibration procedure. Using an adapter, an inclinometer is mounted on the blade. The blade is then rotated at several different pitch angles, and the readings of the Hall sensor output and the inclinometer are recorded. Before the model can be used, a "homing procedure" is performed where the blades are moved to a predefined known position, thereby providing the desired reference.

A third optical encoder is placed on the main shaft for measuring the rotating speed of the rotor and its azimuthal position,

which is necessary for interpreting shaft loads and for performing phase-locked flow measurements. Instead of using a Hall sensor, in this case the calibration is performed manually by placing the rotor at a known azimuthal position.

### 2.5.3 Measurement uncertainty

For every experimental activity it is necessary to estimate the error of the results that it generates. For the tower and shaft loads, given the sensitivity of the strain gauges and the expected strain within the operational regime, the uncertainty is estimated to

be $1\%$. Similarly, the uncertainty of the torque measurement obtained from strain gauges is estimated to range between $2\%$ and $3\%$, depending on the operating point. The manufacturer gives a value of $0.05\%$ for the torque-meter, and below $1\%$ for the Hall sensor. Given the very small dimensions of the collective pitch mechanism assembly and all the uncertainties that this implies, a tolerance of $\pm 0.3°$ can be estimated for the blade pitch angle. Uncertainties in the dimensions of the model (blade length, tower height etc.) and in the measurement of the rotor angular velocity are considered to be negligible.





## 2.6 Control software

The G06 is operated by a Bachmann M1 (Bachmann, 2020) programmable logic controller (PLC), which runs in real time the supervisory logic and the pitch-torque-yaw controllers.

Two analog acquisition modules and one counter module are used for acquiring the sensor readings (strain gauges, encoder), as well as the wind speed. All signals are gathered at a frequency of 250 Hz, except for the torque-meter and shaft bending moments that are sampled at 2.5 kHz. All sensors readings are provided as inputs to the supervisory controller, which is real-time executed by the M1-CPU unit with a clock time of 4 ms; the control pitch, torque and yaw demands are sent to the actuator control boards via a M1-CAN module or by analog output. The real-time controller is organized into several applications written in the C programming language, each handling specialized tasks such as communicating with the actuators, recording data, or calculating actuator demands according to a control algorithm and the state of the machine (idle, power generation etc.).

The control hardware and software is the same for all models of the TUM scaled wind turbine family. Each individual model is uniquely identified by its own ID, which allows the software to select the appropriate model-specific parameters, such as friction tables, controller gains etc. This unified framework simplifies software maintenance and development, and shortens the preparation time for the experimental setup.

## 3 Model characterization

This section presents the basic characteristics of the G06 in terms of its rotor aerodynamic performance, comparing design predictions with measurements obtained in two different wind tunnels. Additionally, the wake is characterized in terms of velocity deficit and wake center deflection in misaligned conditions, and compared to the G1 scaled model and to an engineering wake model. Further results are presented for turbulence intensity (TI), turbulent momentum fluxes and turbulence dissipation rate.

### 3.1 Experimental test conditions

The model was tested in two different atmospheric boundary layer wind tunnels: the BLAST facility at the University of Texas at Dallas (UTD), shown in Fig. 6a, and the tunnel at the Institute of Aerodynamics of the Technical University of Munich (TUM), shown in Fig. 6b.

The UTD BLAST wind tunnel test section has a height of 2.1 m, a width of 2.8 m and a length of 30 m. In this wind tunnel, measurements were taken using Stereo Particle Image Velocimetry (S-PIV) with a LaVision system. The S-PIV equipment comprises of two sCMOS 5.5 Mp cameras mounted on Scheimpflug adapters and equipped with 50 mm Nikon AF 1.8D lenses. A Quantel Evergreen HP laser was used with 380 mJ/pulse, and the cameras were calibrated with a 300 mm by 300 mm dual-plane target. The wake was measured in planes perpendicular to the flow at several downstream distances. All planes





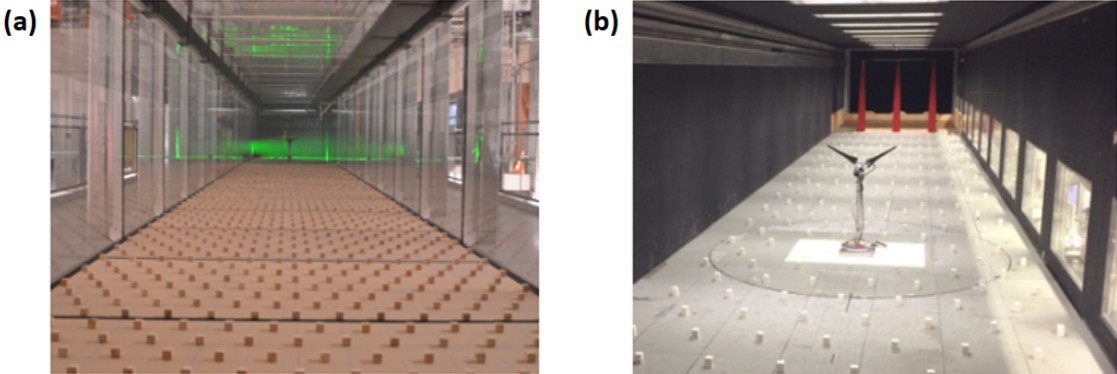

**Figure 6.** The UTD BLAST atmospheric boundary layer test section looking downstream towards the model **(a)**, and the TUM atmospheric boundary layer test section looking upstream **(b)**.

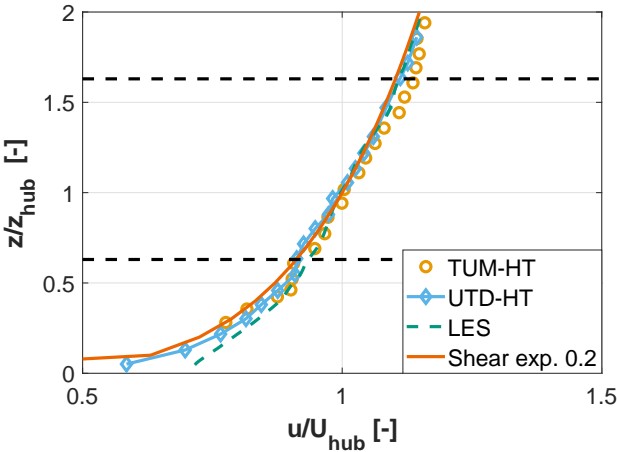

**Figure 7.** Inflow velocity profiles for TUM-HT and UTD-HT inflow conditions; black dashed lines denote the rotor tips.



had a spatial resolution of approximately 0.015 D. The mean flow field for each plane was calculated by averaging 2,000 instantaneous flow fields, which were captured at 10 Hz frequency.

The TUM wind tunnel has a height of 1.8 m, a width of 2.7 m and a length of 27 m. In this wind tunnel the wake was measured using a triple-wire device based on a DISA 55P91 probe and manufactured in-house at TUM (Heckmeier et al., 2019). The three gold-plated tungsten wires have a diameter of 5 $\mu$m with a length of 1.25 mm. The characteristic temperature

coefficient of the sensor is $\alpha_{20} = 0.0036 \, \mathrm{K}^{-1}$. Based on calibration, the overheat ratio, gain, and offset were set to $a_{ov} = 1 : 8$, $G = 2$ and $O = 2$, respectively (Perry and Morrison, 1971).

In both wind tunnels, two different inflow conditions were generated: the first is characterized by a low turbulence and uniform velocity profile, as obtained by the natural development of the flow in the clean wind tunnel; the second was obtained by the use of spires located at the test section inlet and by roughness elements placed on the floor, leading to a higher turbulence

and a sheared velocity profile. The resulting conditions are labelled UTD- or TUM- (depending on the tunnel) LT (for low turbulence) and HT (for high turbulence), and are reported in Table 2, together with the testing conditions in terms of TSR $\lambda$ and thrust coefficient $C_T = T/0.5\rho U^3 \pi R^2$. Figure 7 shows the vertical profile of the normalized streamwise inflow speed $u/U_{\mathrm{hub}}$ for TUM-HT and UTD-HT.

**Table 2.** Summary of test conditions in the UTD and TUM boundary layer wind tunnels.

|  | UTD-HT | UTD-LT | TUM-HT | TUM-LT |
|---|---|---|---|---|
| Cross-section | 2.8 m × 2.1 m | | 2.7 m × 1.8 m | |
| Blockage ratio | 4.8 % | | 5 % | |
| $U_{\mathrm{hub}}$ (wake) | 10.2 ms$^{-1}$ | 10.1 ms$^{-1}$ | 8 ms$^{-1}$ | 10 ms$^{-1}$ |
| $U_{\mathrm{hub}}$ (performance) | – | 5 − 11 ms$^{-1}$ | – | – |
| TI at hub height $I_{\mathrm{hub}}$ | 8.5 % | 0.15 % | 12 % | 0.3 % |
| Shear exponent | 0.2 | 0 | 0.21 | 0 |
| Downstream position ($x$/D) | 2, 3.5, 5, 6.5, 8.5 | | 1, 3, 6 | |
| Wake measurement method | S-PIV | | Hot-Wire | |
| TSRs | $\lambda = 3.5$, $\lambda = 5.35$, $\lambda = 7.2$ | | $\lambda = 7.1$ | |
| Thrust coefficients | $C_T = 0.38$, $C_T = 0.54$, $C_T = 0.72$ | | $C_T = 0.71$ | |

## 3.2 Aerodynamic performance characterization

### 3.2.1 Wind tunnel tests

The aerodynamic performance characterization was performed in the BLAST wind tunnel in UTD-LT conditions (see Table 2). Since blockage is below 5%, no correction was deemed necessary.

Figures 8a-c report the power, thrust and torque coefficients as functions of TSR for several pitch angles. The maximum measured power coefficient is $C_{P_{\max}} \approx 0.41$, which is a good result for such a small rotor, yet 20% lower than the one of the





full-scale reference. The maximum power coefficient is achieved at $\lambda = 7.5$, which is close to the value of 8 of the reference model. However, the difference in performance between $\lambda = 7.5$ and 8 is insignificant due to the flat shape of the curve. At the optimum pitch and TSR, the thrust coefficient is $C_T \approx 0.75$, which is in line with expectations for a full-scale turbine.

Figures 8d-f show the variation of the power $C_P$, thrust $C_T$ and torque $C_Q = C_P/\lambda$ coefficients with respect to TSR for different inflow speeds at a fixed (optimum) pitch angle. The observed dependency of performance on wind speed is relevant

because the G06 turbine is intended for use in waked conditions, where the impinging flow is slower than the free stream. Even though utility scale wind turbines performance coefficients are essentially insensitive to wind speed (except for deformation-induced effects, which however are not present here since the model is rigid), this is not the case for scaled models. Indeed, as seen in the figure, there is an evident performance deterioration as the inflow speed is reduced. This can be explained by the rapid increase in the airfoil drag with decreasing Reynolds number, as shown in Fig. 9. The resulting drop in efficiency

affects primarily the $C_P$ coefficient, as expected, whereas it generates only modest changes in $C_T$, which is mostly driven by lift and not drag. It should be noted that, notwithstanding the reduced and condition-dependent $C_P$, a rotor designed with the criteria adopted here still results in a very realistic wake behavior, as shown later on and more in detailed discussed in Wang et al. (2020a). Additionally, for a wake management application, these characteristics of the power coefficient are not really an issue if the control solution demonstrates improvement over a baseline case. This is in fact one of the roles of scaled models:

although not all physics can always be matched at scale, and therefore absolute values cannot be accurately captured, these models can still typically show trends and changes with respect to a reference (Canet et al., 2021).

Figure 10a shows the variation of power with respect to the yaw misalignment angle $\gamma$, at the optimum pitch angle and tip speed ratio. Fitting the cosine power loss model to the experimental data yields:

$$P = P_{\gamma=0} \left( \cos \gamma \right)^{2.01}. \tag{5}$$

The power loss exponent for the G1 scaled wind turbine is 2.17 (Campagnolo et al., 2020), while Pedersen (2004) reported 2, Schepers (2001) 1.8, and Damiani et al. (2018) found 1.9. Other studies have found values closer to the theoretical limit of 3 (Bastankhah and Porté-Agel, 2015; Bartl et al., 2018).

Figure 10b shows the $C_P$-$\lambda$ curves of two different G06 rotors for the same TUM-HT inflow conditions and a blade pitch angle $\beta = 0°$. Results indicate that the two rotors have an almost identical performance, which validates the repeatability of

the manufacturing, calibration and data acquisition processes.

### 3.2.2  Numerical simulations: polar identification

One of the intended uses of the G06 turbine is the validation of simulation tools. Most numerical models of rotor aerodynamics depend on the airfoil lift and drag coefficients (polars). Especially for scaled models, the determination of the airfoil polars involves considerable uncertainties. In fact, manufacturing imprecisions, in combination with the small dimensions of the

blade, can have significant effects on the airfoil shape and, consequently, on its polars. As a result, the nominal polars used for designing the rotor might not be completely accurate.



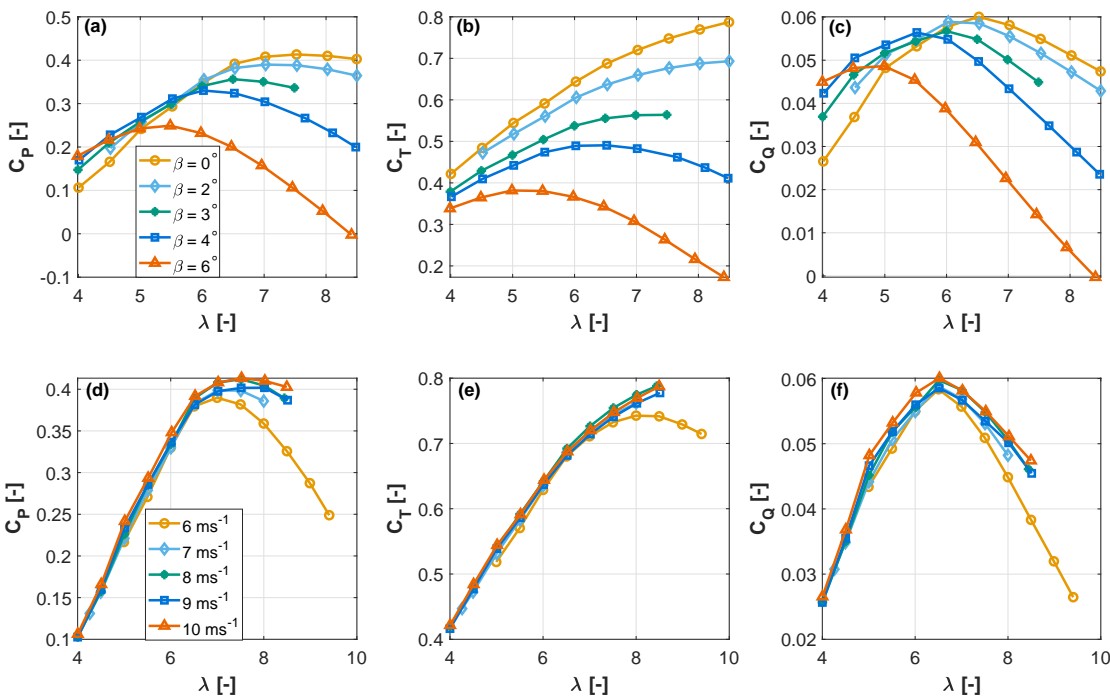

**Figure 8.** Power $C_P$ **(a,d)**, thrust $C_T$ **(b,e)** and torque $C_Q$ **(c,f)** coefficients as functions of TSR $\lambda$ for different pitch angles $\beta$ at 10 ms$^{-1}$ **(a,b,c)**, and for different wind speeds at the optimum pitch angle **(d,e,f)**.

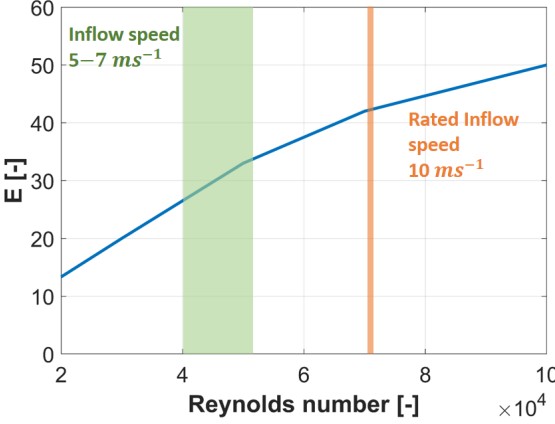

**Figure 9.** Efficiency of the RG-14 airfoil as a function of Reynolds numbers, as computed with Xfoil (Drela). The orange line indicates the G06 Reynolds operating regime at rated speed. The green area indicates the approximate Reynolds number of waked wind turbines that are further downstream in a column configuration.





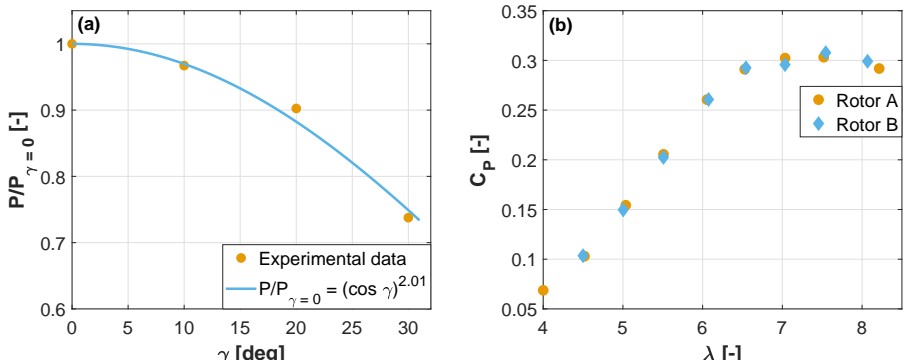

**Figure 10.** Power output as a function of the wind misalignment angle $\gamma$, normalized with respect to the $\gamma = 0°$ case **(a)**. $C_P$ vs. $\lambda$ for two G06 rotors in the same inflow conditions (TUM-HT) and same pitch angle $\beta = 0°$ **(b)**.

To address this problem, Bottasso et al. (2014a) developed a method for identifying the airfoil aerodynamic characteristics directly from measurements of the power and thrust produced by the rotor. By this method, the nominal polars are corrected, resulting in tuned aerodynamic characteristics that better reflect the actual conditions on the manufactured rotor. This maximum-likelihood calibration procedure was further improved in Wang et al. (2020b), to better account for measurement errors.

This method was used here to tune the polars, using 160 different operating conditions measured in the UTD wind tunnel in UTD-LT inflow. Figure 11a shows the airfoil efficiency as a function of angle of attack for the nominal and tuned polars. Results show that, although not identical, the difference between the two sets of polars is small, which seems to indicate a good overall manufacturing precision of the blades. This small difference has also a relatively small effect on the circulation distribution, as shown in Fig. 11b. This same figure also reports the normalized circulation distribution of the reference model obtained with FAST (Jonkman and Jonkman, 2018). Results show that, outboard of $r/R = 0.3$, the circulation of the G06 blade is almost identical to the reference one when using the nominal polars; this is expected, as this condition is explicitly enforced in the rotor design problem (see Eq. (3)). When considering the identified polars, the circulation matching error is less than 2%, which is a more than satisfactory result given the small size of the rotor. The difference between the G06 and reference circulations in the innermost 30% of blade span is due to the rather long extent of the cylindrical root of the scaled blade, due to manufacturing reasons.

### 3.3 Wake characterization

#### 3.3.1 Velocity deficit, recovery and wake deflection

This section aims at characterizing the wake of the G06 turbine in terms of velocity deficit, recovery rate, and path deflection as a function of misalignment angle.





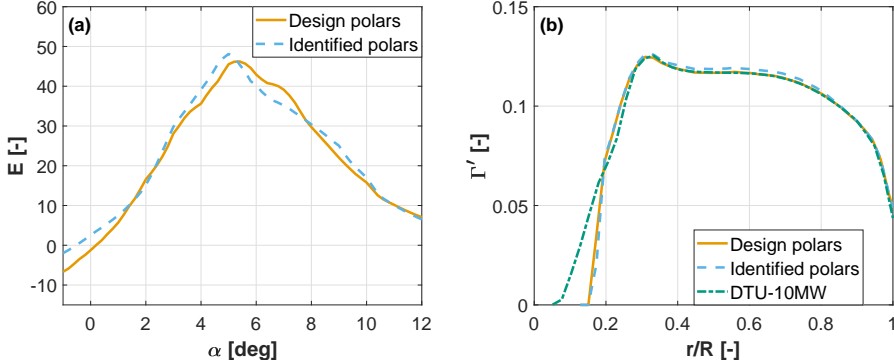

**Figure 11.** Comparison between airfoil efficiency $E$ calculated with the nominal polars and the identified ones, for a chord-based Reynolds equal to 70,000 **(a)**. Nondimensional circulation distribution $\Gamma'$ along the blade span $r/R$ for the G06 using the nominal design and the identified polars, and for the reference turbine **(b)**.

Considering the number of parameters that can affect the results, the repeatability of wake measurements was verified in different wind tunnels and with different measurement techniques. To this end, the turbine wake was measured at different downstream distances in the UTD wind tunnel in UTD-LT conditions using S-PIV, and in the TUM wind tunnel in the compa-

rable TUM-LT inflow using hot-wire probes. Figure 12 shows an excerpt from this data set, reporting both the lateral (panel a) and vertical (panel b) wake profiles obtained at $x/D = 3.5$. Results show a very good agreement between the two measurements, with an average error of 1.5% and a standard deviation of 1%. Similar results, not shown here for brevity, were obtained at other downstream distances. The good match between these two sets of measurements serves as an additional validation of the calibration, measurement and postprocessing procedures.

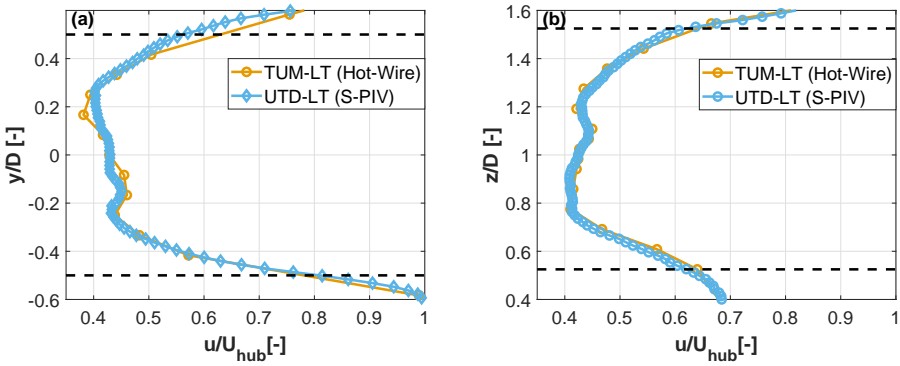

**Figure 12.** Comparison of wake measurements at $x/D = 3.5$ in two different wind tunnels and with two different measurement techniques. The comparison is made for laminar and uniform inflow (UTD-LT, TUM-LT). Black dashed lines indicate the rotor tips. Horizontal profile **(a)**; vertical profile **(b)**.





Figure 13 reports horizontal and vertical profiles of normalized velocity deficit for the laminar and uniform TUM-LT and
sheared and turbulent TUM-HT conditions. Results for the TUM-LT inflow conditions reveal, especially for the horizontal
scan, the typical double-Gaussian profile in the near wake (Schreiber et al., 2020a). As expected, in the TUM-HT case the
higher TI accelerates the dissipation of the nacelle wake, resulting in a single-Gaussian profile (Bastankhah and Porté-Agel,
2017a; Vermeer et al., 2003). The vertical profile is distorted by the presence of the boundary layer in the TUM-HT inflow
case. The profiles of the two different inflow conditions are similar immediately behind the rotor at $x/D = 1$, where recovery
has not yet initiated and the deficit is mainly driven by the extraction of kinetic energy from the flow performed by the wind
turbine. On the other hand, the evolution further downstream is markedly different, on account of the different TI.

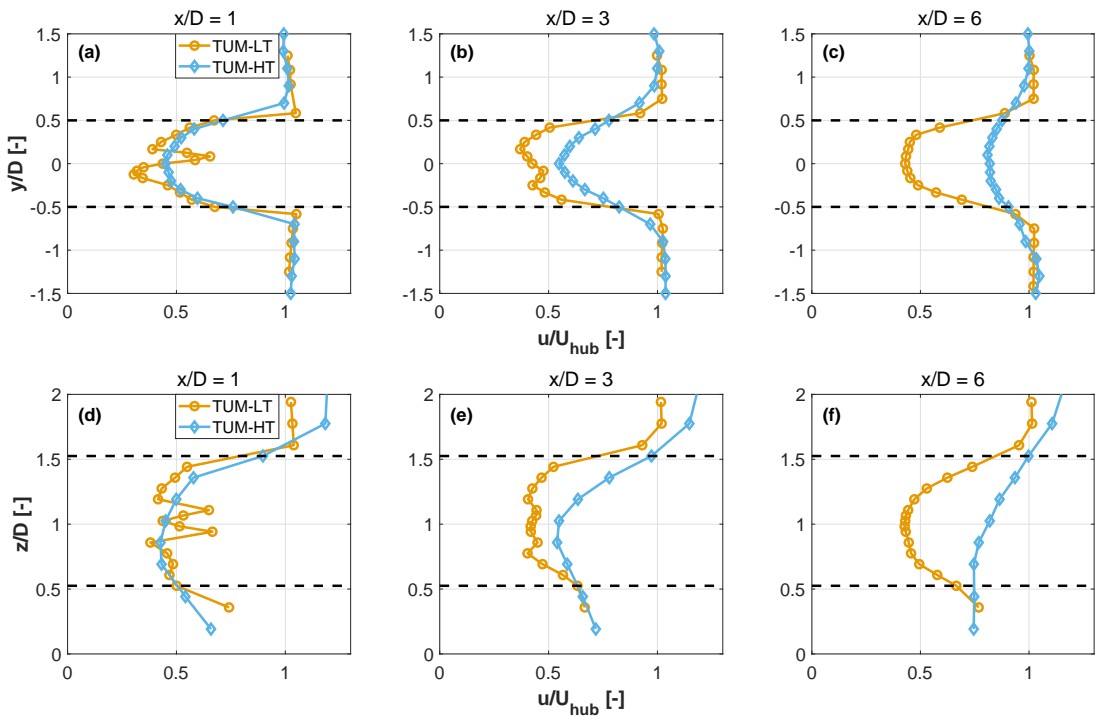

**Figure 13.** Horizontal **(a-c)** and vertical **(d-f)** profiles of the normalized streamwise velocity at several downstream distances, for sheared
turbulent (TUM-HT), and uniform laminar (TUM-LT) inflow conditions. Black dashed lines indicate the rotor tips.

    Figure 14 shows the downstream evolution of the velocity deficit at wake center for different thrust coefficients in UTD-HT
inflow. The experimental data is plotted together with the predictions of the model of Bastankhah and Porté-Agel (2014). The
model depends on the thrust coefficient and a wake growth parameter, which was calculated according to Cheng and Porté-Agel
(2018). Results show that experimental data and model predictions are in good agreement, with the exception of the low-thrust
cases ($C_T = 0.38$, $C_T = 0.54$) closer to the rotor disk (up to $x/D = 3.5$), where the model overpredicts the wake velocity. This
is probably due to the wake of the nacelle still being a contributing factor at this distance and position. The figure also clearly





shows that lower thrust coefficients are associated with slower recovery rates, which partially explain why static derating wind
farm control strategies lead to only limited power gains (Annoni et al., 2016; Campagnolo et al., 2016a).

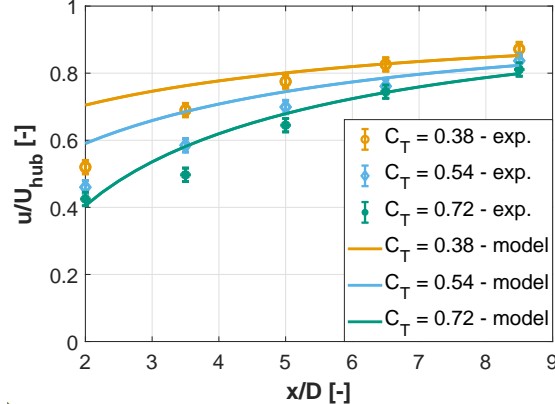

**Figure 14.** Velocity deficit evolution in UTD-HT inflow, for several values of $C_T$, against the wake model of Bastankhah and Porté-Agel (2014).

The wake of the G06 was also compared to the one of the G1 model, a scaled turbine designed using similar criteria and already extensively used for wake and wind farm control studies (Campagnolo et al., 2016b; Schreiber et al., 2017b; Wang et al., 2019, 2020a; Bottasso and Campagnolo, 2020; Campagnolo et al., 2020). Figure 15 shows lateral profiles of normalized streamwise velocity at hub height 5D downstream of the two turbines. The profiles are compared in the wind-aligned condition
$\gamma = 0°$, and for a high misalignment angle of $\gamma = 30°$. The G06 model operates in the UTD tunnel in HT conditions at a thrust coefficient $C_T = 0.72$, and the speed profile was obtained from S-PIV measurements. The G1 was tested in the wind tunnel at Politecnico di Milano in a condition characterized by a vertical shear of 0.2, a TI of 10% and $C_T = 0.75$, and the wake profile was measured with triple hot-wire probes. Notwithstanding the different models, wind tunnels and measurement techniques, the wake profiles both in aligned and misaligned conditions are in good agreement with each other.
Finally, following Wang et al. (2020a), the wake of the G06 was compared to the one of its reference, to verify to what extent the scaled wake represents the characteristics of its full-scale counterpart. To this end, simulations were conducted with the large-eddy simulation (LES) actuator-line method (ALM) implemented in the flow solver described by Wang et al. (2019), and already validated in previous work. To ensure a meaningful comparison, the scaled and full-scale models were simulated with the same code, using exactly the same numerical methods and algorithmic parameters. Specifically, the fluid grid and
the ALM discretization were scaled up according to the geometric scaling factor, whereas all other numerical and algorithmic parameters of the solver were kept exactly the same for the scaled and full-scale simulations. The two wind turbine models were also exposed to the same identical ambient turbulent inflows at their respective scales. To achieve this result, first the G06 inflow was obtained by simulating the UTD wind tunnel test section to match the UTD-HT conditions (see Fig. 7); next,



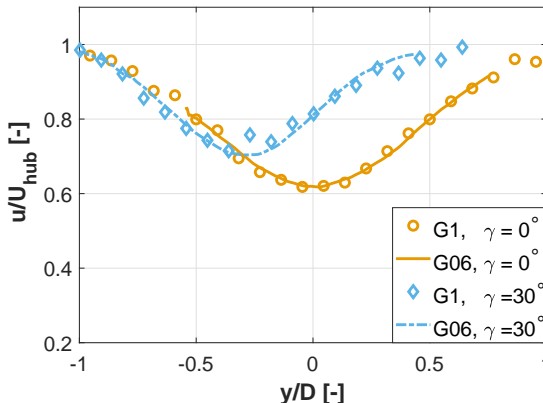

**Figure 15.** Velocity deficit for the G06 and G1 turbines 5D downstream of the rotor, for both a wind-aligned and a high misalignment angle of $\gamma = 30°$. Measurements were taken at similar thrust coefficients; $C_T = 0.72$ for G06 and $C_T = 0.75$ for G1, in similar turbulent inflows in different wind tunnels.

the DTU 10 MW inflow was generated by scaling up the G06 one based on the time and length scaling factors, following the
approach described in Wang et al. (2020a).

Figure 16 shows contours (looking upstream) of the normalized streamwise velocity difference in the wakes of the G06 and of the DTU 10 MW reference, computed as

$$\frac{(u/U_{\text{hub}})_{\text{G06}} - (u/U_{\text{hub}})_{\text{DTU}}}{(u/U_{\text{hub}})_{\text{DTU}}}, \tag{6}$$

where the subscripts $(\cdot)_{\text{G06}}$ and $(\cdot)_{\text{DTU}}$ stand for the respective turbines; in the same figure, the arrows indicate the difference
in the normalized in-plane velocity components. The comparison is made at two downstream distances, namely immediately behind the rotor disk at $x/D = 1$ (panel a) and at $x/D = 5$ (panel b).

To isolate the effects due to the rotor, the turbine tower and nacelle were not included in the simulations. The models were operating at their respective optimum pitch angle and at TSR $\lambda = 8$. In these conditions the G06 has a $C_P = 0.41$ and a $C_T = 0.75$, whereas the full-scale turbine has a $C_P = 0.47$ and a $C_T = 0.81$.
The figure indicates that at $x/D = 1$ the G06 wake speed is faster on a ring that covers approximately 50% of the blade span, on account of the lower $C_T$. There is also a difference at the center of the wake because of the larger hub diameter of the G06 (see Fig. 11b). The counterclockwise rotation of the in-plane velocity difference indicates a stronger swirl of the DTU 10 MW wake, because of its higher $C_Q$. Notwithstanding these differences immediately behind the rotor, at $x/D = 5$ the wakes appear to be very similar, with errors in the longitudinal speed component around $1-2\%$ for most of the domain, reaching a maximum
of $3\%$ in the center of the wake. At this distance the wake rotation has dissipated almost completely, and the in-plane velocity vectors have been removed from the figure.



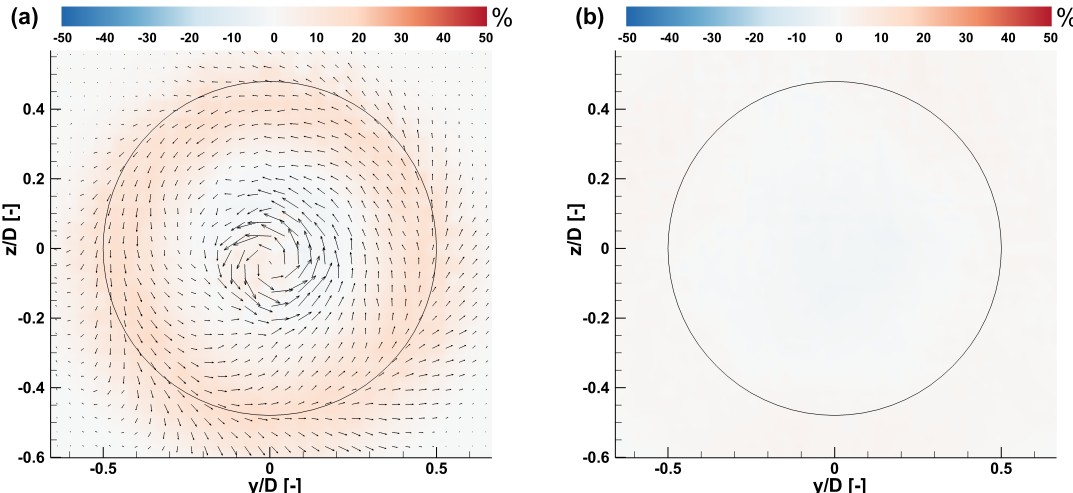

**Figure 16.** Normalized streamwise velocity difference in the wakes of the G06 of the DTU 10 MW at $x/D = 1$ **(a)** and at $x/D = 5$ **(b)**. The black circle denotes the rotor circumference, while arrows indicate the difference between in plane velocities.

Wang et al. (2020a) presents a more comprehensive discussion on the comparison of full-scaled and scaled wakes, considering the G1 turbine. That study shows that a scaled rotor —designed according to the principles followed also here for the G06— generates wakes that are in very good agreement with full-scale ones with respect to a number of different metrics.

### 3.3.2 Turbulence intensity

Within the wake of a wind turbine, the TI level is typically different than the ambient one. In fact, additional turbulence is produced by the boundary layers forming on the rotor blades, by the flow that separates from the tower and the nacelle, and by the velocity shear within the wake (Quarton and Ainslie, 1990). The so-called "added" TI (Ainslie, 1986) is used to quantify the change in turbulence with respect to the ambient conditions, and it is defined as

$$I_{\mathrm{add}} = +\sqrt{I^2 - I_{\mathrm{hub}}^2}, \qquad I \geq I_{\mathrm{hub}}, \tag{7a}$$

$$I_{\mathrm{add}} = -\sqrt{I_{\mathrm{hub}}^2 - I^2}, \qquad I < I_{\mathrm{hub}}, \tag{7b}$$

where $I$ is the TI at a generic point, while $I_{\mathrm{hub}}$ is the TI at hub height.

Figure 17 shows contour lines of $I_{\mathrm{add}}$ in UTD-HT inflow at several downstream distances in aligned conditions for $C_T = 0.72$, as obtained from the post-processing of S-PIV measurements in the UTD tunnel. The figures show that the influence of the rotor on the flow is highly nonuniform. In fact, the added TI has a horseshoe shape with a maximum at the top of the rotor; this region of higher TI is sharp and highly localized immediately behind the rotor and diffuses moving downstream. The lower-center part of the wake is characterized by an added TI that is either negligible or slightly negative, i.e. lower than the ambient one. This effect could have the following exegesis: due to the presence of the boundary layer, the velocity deficit





induced by the rotor results in an increased vertical shear in the top part of the wake, whereas a decreased vertical shear is
generated at the bottom of it (see also the vertical speed profiles in Fig. 13). Therefore, the reduced —with respect to the
ambient condition— vertical shear in the lower part of the wake results in a reduction of turbulence intensity. Similar results
have been reported by Bastankhah and Porté-Agel (2017c).

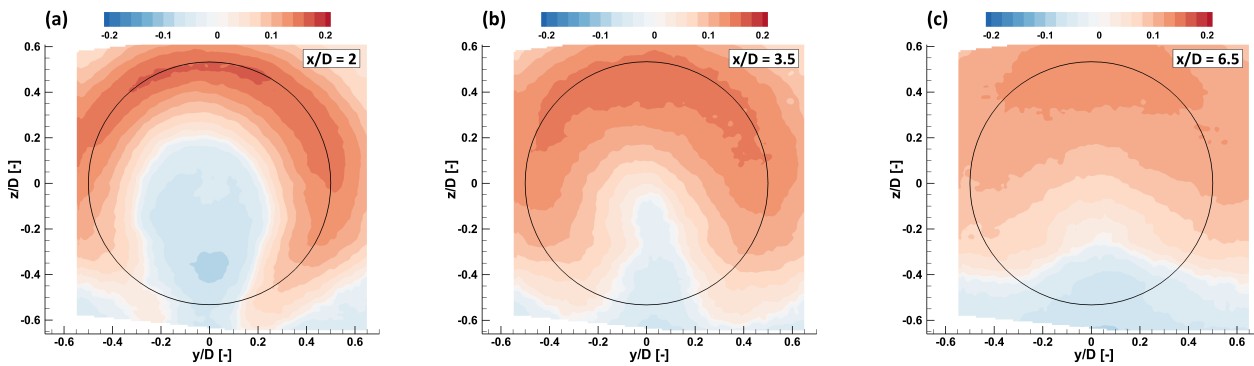

**Figure 17.** Added turbulence intensity $I_{\mathrm{add}}$ for UTD-HT inflow at several downstream distances. The black circle denotes the rotor circumference.

Figure 18 shows vertical and lateral profiles of added TI in high turbulence sheared inflow (TUM-HT) and laminar uniform
inflow (TUM-LT), for a wind aligned condition at $C_T = 0.72$. These results are coherent with the ones of the previous figures,
and show that for the sheared inflow the maximum added TI is found at the top of the rotor disk, whereas at the center and
bottom the values are slightly negative and reduce in magnitude while moving downstream. For the uniform inflow case, the
profiles are nearly symmetrical, with a markedly slower evolution on account of the weak mixing; the nacelle wake effects are
also clearly visible in the immediate vicinity of the rotor.

Several studies have considered the modelling of added TI, because of its importance in wake recovery and in the loading
experienced by downstream machines. Figure 19 shows a comparison between experimental data for the G06 in UTD-HT
inflow and the empirical model for the maximum added TI proposed by Crespo and Hernández (1996). This empirical model
is applicable beyond 5D downstream of the rotor, and it writes:

$$I_{\mathrm{add_{max}}} = 0.73 a^{0.8325} I_{\mathrm{hub}}^{0.0325} (x/D), \tag{8}$$

where $a$ is the axial induction factor. The figure shows that there is a very good agreement between the estimated and the mea-
sured maximum added TI. This provides an additional confirmation of the realistic behavior of the wake even from this point
of view, since this model has been verified against numerical simulations and field data at full scale (Crespo and Hernández,
1996; Niayifar and Porté-Agel, 2015).

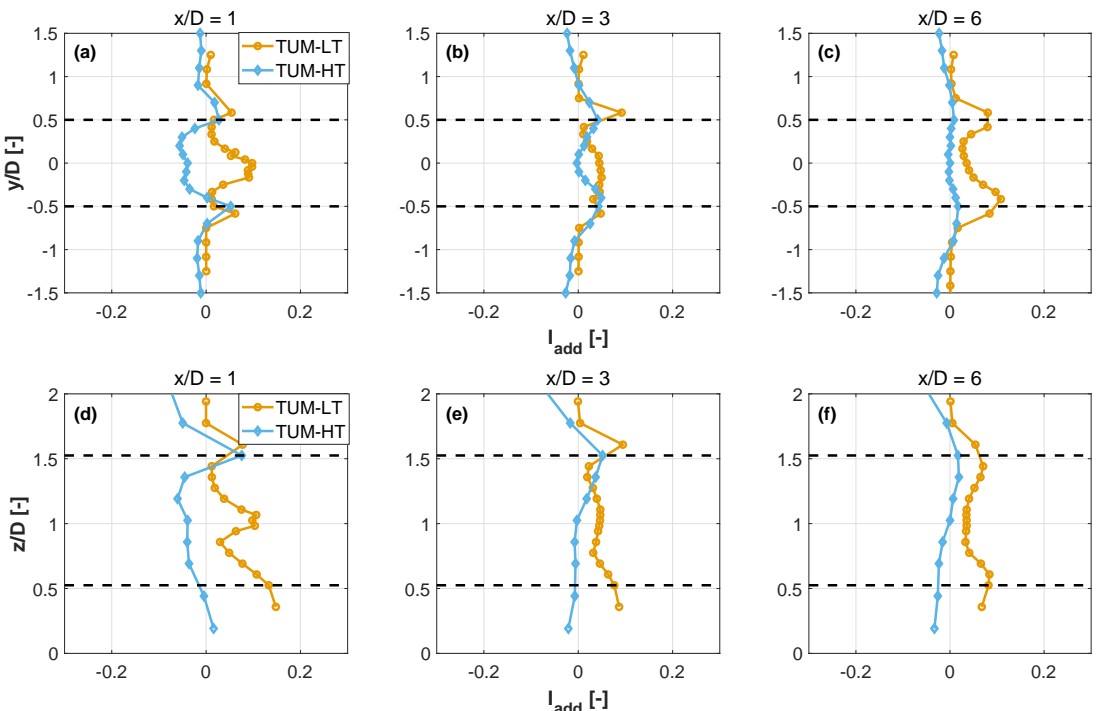

**Figure 18.** Horizontal **(a-c)** and vertical **(d-f)** profiles of added TI at several downstream distances and different inflow conditions. Black dashed lines indicate the rotor tips.

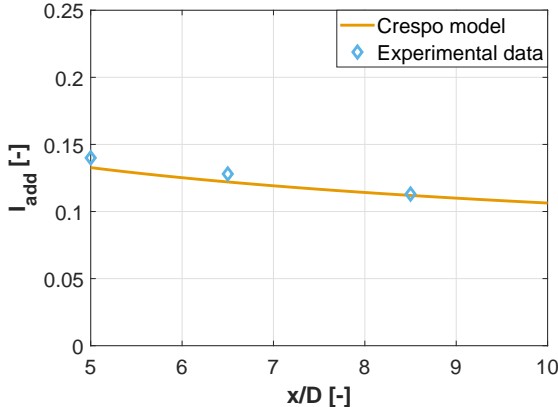

**Figure 19.** Maximum added TI vs. downstream distance, for the G06 in UTD-HT inflow and the the empirical model of Crespo and Hernández (1996).





### 3.3.3 Turbulent momentum fluxes

After Reynolds decomposition and time averaging (Durst, 2008), the momentum equation reads:

$$\overline{\rho u_i}\frac{\partial \overline{u_j}}{\partial x_i} = -\frac{\partial \overline{p}}{\partial x_j} + \frac{\partial}{\partial x_i}\underbrace{\left(\mu\frac{\partial \overline{u_j}}{\partial x_i} - \overline{\rho u_i' u_j'}\right)}_{\tau_{ij}} + \overline{\rho}g_j, \tag{9}$$

where $u$ is velocity, $t$ is time, $x$ is a spatial coordinate, $p$ is pressure and $\mu$ is kinematic viscosity. The subscript $(\cdot)_i$ refers to a component in a Cartesian coordinate system, while $(\cdot)'$ and $\overline{(\cdot)}$ denote the fluctuating and time-averaged values of the relevant quantities, respectively. The Reynolds decomposition introduces additional terms to the molecular momentum transport equation, which represent turbulent velocity fluctuations $\overline{\rho u_i' u_j'}$ for $i \neq j$ and are called turbulent momentum fluxes (or Reynolds stresses). These terms express the main mechanism of re-energization of the wake, as they are responsible for entraining ambient high-momentum flow into it.

Figure 20 shows contours of the normalized lateral turbulent momentum flux $\overline{u'w'}/U_{hub}^2$, while Fig. 21 shows contours of the normalized vertical flux component $\overline{v'w'}/U_{hub}^2$. Measurements were obtained with sPIV in UTD-HT inflow conditions at a thrust coefficient $C_T = 0.72$. Qualitatively, the figures show that the exchange of momentum due to turbulent velocity fluctuations increases moving downstream (compare the figures at $x/D = 2$ and $x/D = 3.5$), reaching deeper into the wake core. This is in agreement with previous studies (Bastankhah and Porté-Agel, 2017a) and in line with the observation that the breakdown of the tip vortices, which occurs at approximately $x/D = 4$, removes a separation layer between the wake and the ambient flow, thereby facilitating the exchange of momentum (Medici, 2006).

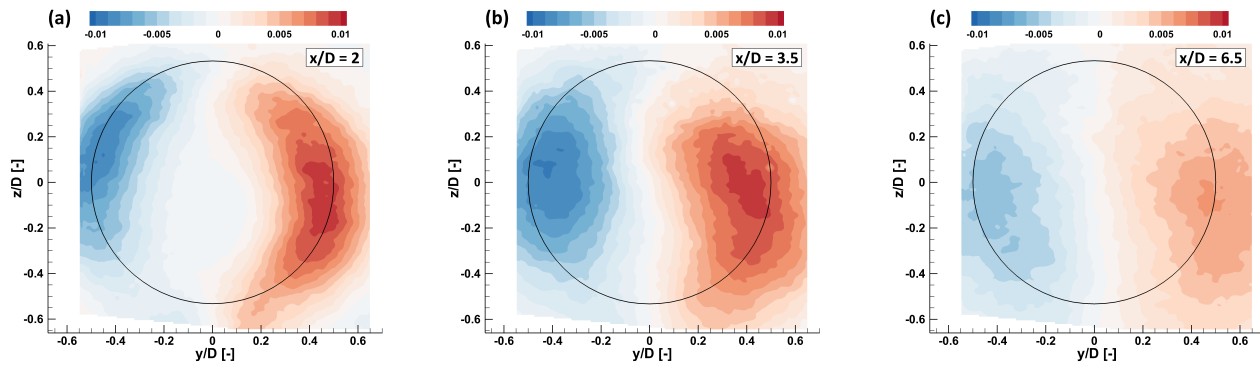

**Figure 20.** Normalized lateral turbulent flux $-\overline{v'u'}/U_{hub}^2$ for UTD-HT inflow at several downstream distances. The black circle denotes the rotor circumference.

Figure 22 shows profiles of lateral and vertical turbulent momentum fluxes at different downstream positions and for different thrust coefficients, in the same UTD-HT inflow. The figure shows that a higher thrust coefficient leads to stronger turbulent momentum fluxes. The figure also allows one to appreciate how the vertical momentum flux dissipates quickly in the lower part of the rotor disk, a result of the reduced shear shown in Fig. 13d-f. The lack of symmetry for both the lateral and vertical





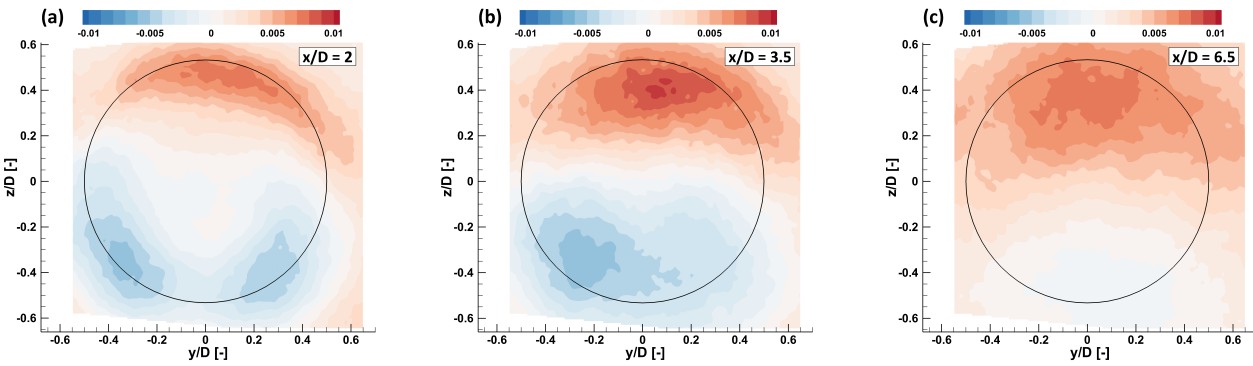

**Figure 21.** Normalized vertical turbulent flux $-\overline{w'u'}/U_{hub}^2$ for UTD-HT inflow at several downstream distances. The black circle denotes the rotor circumference.

turbulent fluxes is probably related to the rotating motion of the wake (Chamorro and Porté-Agel, 2009). Furthermore, it appears that the lateral momentum flux maximum value is higher than the vertical one at any position, similarly to the results
obtained in wind tunnel tests by Bastankhah and Porté-Agel (2017c) and by CFD simulations by Shamsoddin and Porté-Agel (2016), on account of the more pronounced lateral than vertical meandering (Bastankhah and Porté-Agel, 2017a).

### 3.3.4   Dissipation rate

The analysis of the turbulent energy budget provides further insight into wake behavior. The kinetic energy equation for the turbulent flow is derived from the momentum equation after averaging over time and subtracting the energy equation of the
mean flow, which results into the expression

$$
\underbrace{\overline{\rho u_i} \frac{\partial}{\partial x_i}\left(\frac{1}{2}\overline{u_j'^1}\right)}_{\frac{\partial k}{\partial x_i}} = \underbrace{-\frac{\partial}{\partial x_j}\left(\overline{p'u_j'}\right) + \frac{\partial}{\partial x_j}\left(\mu \overline{u_j'\frac{\partial u_j'}{\partial x_i}}\right) - \frac{\overline{\rho}}{2}\frac{\partial}{\partial x_i}\left(\overline{u_i'u_j'^2}\right)}_{\frac{\partial D_{kj}}{\partial x_j}} \underbrace{-\overline{\rho u_i'u_j'}\frac{\partial \overline{u}_j}{\partial x_i}}_{P_\kappa} \underbrace{-\mu \overline{\frac{\partial u_j'}{\partial x_i}\frac{\partial u_j'}{\partial x_i}}}_{\epsilon_\kappa},
\tag{10}
$$

where $k$ is the turbulent kinetic energy and $D_k$, $P_\kappa$ and $\epsilon_\kappa$ are the turbulent kinetic energy diffusion, production, and dissipation, respectively. This last term represents the rate at which turbulent kinetic energy is transformed into heat, and it is an important parameter for the evolution of the wake.
Despite its relevance, only a few studies report an analysis of the dissipation rate of wind turbine wakes: Smalikho et al. (2013) and Lundquist and Bariteau (2015) analyzed data from field experiments, while Hamilton et al. (2012) calculated the dissipation rate in a scaled wind farm employing hot wire anemometry with a high sampling frequency of $40$ kHz. In fact, the dissipation rate of turbulent kinetic energy can be directly calculated from experimental data, provided that the sampling frequency is sufficiently high to capture the smallest eddies in the flow. If this requirement is not fulfilled, the inertial dissipation
approach can be employed (Champagne, 1978). This method is based on the inertial subrange theory, which suggests that the rate of energy transfer from bigger eddies to medium size eddies is equal to the dissipation rate of the smallest eddies in the

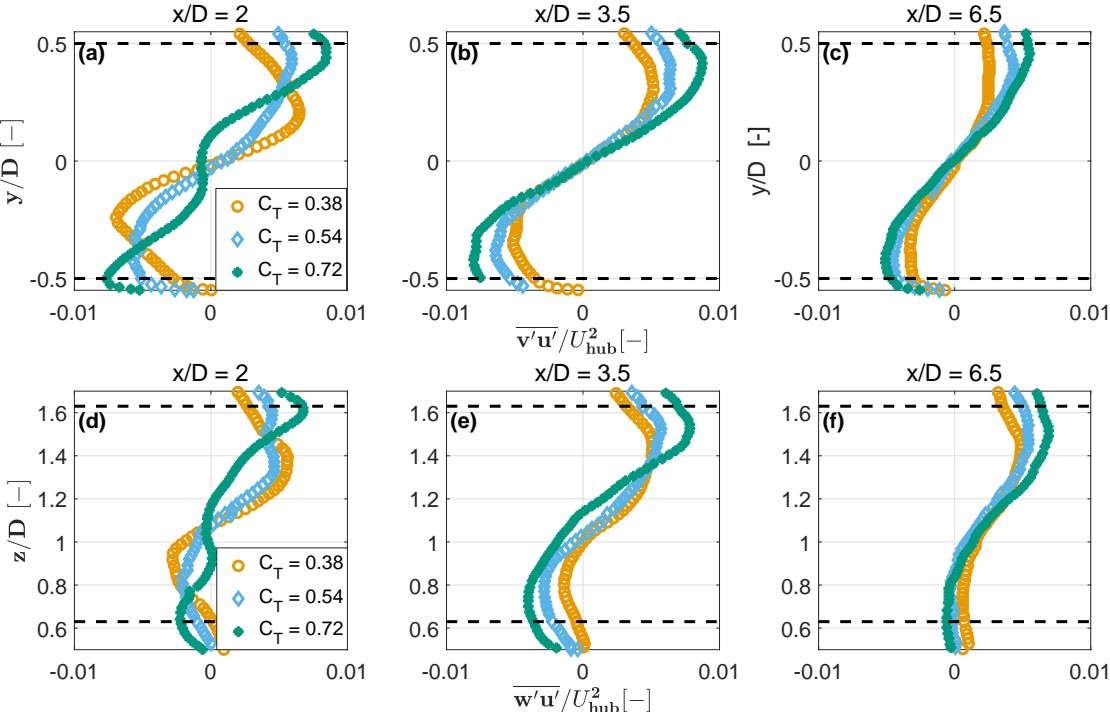

**Figure 22.** Horizontal **(a-c)** and vertical **(d-f)** profiles of the normalized lateral and vertical turbulent fluxes at several downstream distances and for different thrust coefficients, in the sheared turbulent UTD-HT inflow. Black dashed lines indicate the rotor tips.

energy cascade. Therefore, a sensor that is capable of capturing the inertial subrange of the energy cascade is also adequate for calculating the dissipation rate according to the following formula:

$$\epsilon_\kappa = \left(\frac{2\pi}{U}\right)\left(\frac{f^{5/3} S_u(f)}{k}\right)^{3/2},$$  (11)

where $S_u(f)$ is the power spectrum of the velocity $u$ in the inertial subrange, while $f$ is frequency and $k = 0.52$ is the Kolmogorov constant (Fairall and Larsen, 1986; Lundquist and Bariteau, 2015). The inertial subrange can be estimated from the fast Fourier transform of the $u$ velocity. Next, the average value of $f^{5/3} S_u(f)$ can be computed over this frequency band. This same approach was used here.

Figures 23a and 23b show, respectively, the horizontal and vertical profiles of the dissipation rate at different downstream 540 distances, for TUM-HT inflow conditions. A qualitative analysis of the results shows that the dissipation rate inside the wake is almost two orders of magnitude higher than in the ambient flow, which agrees with the observations of Lundquist and Bariteau (2015). Moreover, the dissipation rate profiles have a similar shape to the added TI ones (see Fig. 18). Even though the sampling frequency requirements suggested in the literature are met here, the accurate quantification of the dissipation rate was a rather tedious procedure with a considerable degree of uncertainty, similarly to what reported in Bluteau et al. (2011). The





main sources of uncertainty are the estimation of the inertial subrange frequency band and the assumption of the Kolmogorov constant, in addition to important factors in the calculation of the dissipation rate —such as flow characteristics (anisotropy, shear etc.) and instrumentation limitations (signal to noise ratio, sampling frequency).

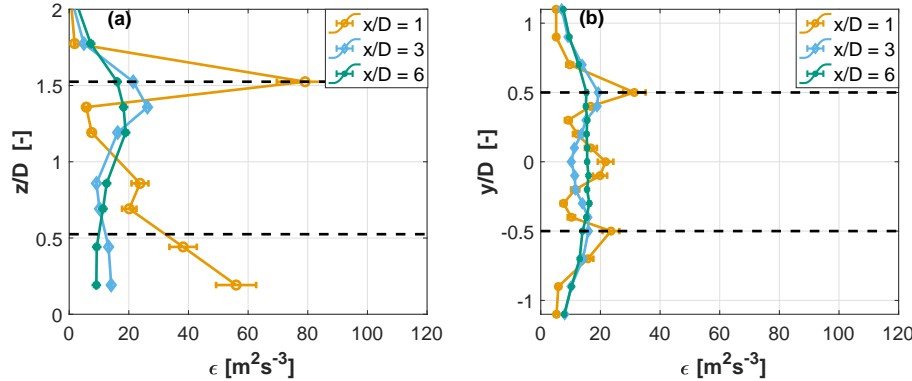

**Figure 23.** Horizontal **(a)** and vertical **(b)** profiles of the dissipation rate $\epsilon_\kappa$ at several downstream distances, for TUM-HT inflow conditions. Black dashed lines indicate the rotor tips.

The uncertainty in the inertial subrange was estimated following Piper (2001), and reported in Figs. 23 in the form of error bars. Results indicate that the error in $\epsilon_\kappa$ is around $10\%$ at $x/D = 1$, which however diminishes considerably when

moving further downstream. Sreenivasan (1995) reviewed hundreds of experiments, considering different flows and different applications, and concluded that approximately the same value of the Kolmogorov constant applies to all those conditions. More specifically, for isotropic flows, the constant was found to have a mean value of 0.53 with a standard deviation of $10\%$. Given that the Kolmogorov constant appears in Eq. (11) to the power of $3/2$, a $10\%$ deviation in the constant leads to a $15\%$ deviation in the dissipation rate.

**4   Conclusions**

This paper has presented the design and characterization of the new scaled multipurpose wind turbine model G06. The need to design the G06 arose from an increased interest in the understanding of plant and complex-terrain flows, including improved operation by wind farm control. In fact, given the challenges posed by full-scale field measurements, experiments conducted in boundary layer wind tunnels with sophisticated small-scale wind turbines are attracting an increased attention from the

research community and are providing additional opportunities for the collection of high-quality data sets. The characterization of the model served the purposes of verifying that the turbine operates as intended, and represented an opportunity to generate reference measurements to support future studies.

The foreseen use cases demand close-loop controls and sensorization in a compact size, yet with realistic aerodynamic characteristics, including at the rotor and in the near- and far-wake regions. The blade was designed to match the circulation



distribution of a full-scale reference at the same optimum TSR. Effects caused by the unmatched chord-based Reynolds number were mitigated by the use of an ad hoc airfoil. To evaluate the as-manufactured performance of the blades, the airfoil polars were identified directly from rotor power and thrust measurements using a dedicated estimation procedure. The identified polars are only marginally different from the nominal ones, resulting in a very good quality match of the circulation distribution over the outboard 75% of the blade span. High fidelity LES-ALM simulations of the G06 and its full-scale reference showed a very good agreement between the two wakes, resulting in errors of a few percent points in the streamwise velocity component of the developed far wake; additionally, the two turbines have an almost identical thrust coefficients at the design TSR. Lastly, the comparison between two different G06 rotors achieved extremely similar characteristics, demonstrating the repeatability and consistency of the manufacturing, calibration and measuring procedures.

The G06 wake was extensively tested in two different boundary layer wind tunnels and two different inflows, a laminar one and a sheared turbulent one. The measurements in both wind tunnels revealed the expected strong influence of inflow conditions on the wake profiles and recovery rate. Comparisons with the G1 turbine and with an engineering wake model showed very good agreement, both in terms of velocity deficit within the wake and wake deflection in yaw misaligned conditions.

The wind tunnel data was also used to analyze high-order flow statistics, including added TI, turbulent momentum fluxes and turbulence dissipation rate. Contour plots of the added TI revealed a horseshoe shape, with a maximum in the upper wake region and small or negative values in the center-lower region. Comparison of the measured maximum added TI with the Crespo and Hernandez empirical model showed a very good agreement.

Profiles of the turbulent momentum fluxes showed that higher thrust coefficients lead to a higher transfer of momentum flux from the ambient flow inside the wake, leading to a faster wake recovery. The turbulent momentum fluxes reach a maximum at $x/D = 3.5$, where also the fastest speed recovery is found, probably on account of the vortex breakdown taking place in this region of the wake.

The turbulence dissipation rate was also characterized in this work, for the first time directly from wind tunnel measurements. It was found that the inertial dissipation method poses challenges in the accurate estimation of the inertial subrange frequency band and the Kolmogorov constant. Nevertheless, the resulting shape of the profiles were found to be rather insensitive to the uncertainties, and were also in line with similar field measurements at full scale.

The characterization conducted so far seems to indicate that the new scaled G06 turbine satisfies the initial requirements, works reliably without any evident weakness, and is ready for supporting future wind tunnel test campaigns. Undoubtedly, the turbine can be further improved and several of the topics addressed in this paper can be analyzed in greater depth. On the hardware side, a second generation of the turbine could include individual pitch control, for example by using a swashplate, and simplifications in the wiring, for example eliminating the slip ring in favour of wireless technology. Faster, simpler and even more precise manufacturing of the blades could be obtained by 3D printing. Regarding capabilities, the wind observation technology of Schreiber et al. (2018, 2020b) has still to be demonstrated and validated on the G06, in support of advanced wind farm control strategies. Finally, the fidelity of the wake of the G06 with respect to the full-scale reference should be more extensively verified, following the approach of Wang et al. (2020a) and even using higher fidelity CFD simulations. In fact, a



thorough understanding of the fidelity and limits of this —and in general of all— scaled models is of crucial importance, for a
correct interpretation of the results and their scientific credibility.

*Data availability.*    Data from the experiments is available upon request.

*Author contributions.*    EMN designed, assembled and operated the G06 turbine, performed the wind tunnel experiments at TUM and an-
alyzed the results; CLB defined the design requirements, defined the design methods, contributed to the interpretation of the results and
supervised the whole work; EMN and CLB wrote the manuscript; FC contributed to the design of the G06, developed the rotor design code,
and performed the wind tunnel measurements with the G1 turbine. EMN and SL conducted the experiments at UTD; VGI supervised the
experiments at UTD, and contributed to the interpretation of the results; MAR facilitated and supported the experiments at UTD. All authors
provided important input to this research work through discussions, feedback and by improving the manuscript.

*Competing interests.*    The authors declare that they have no conflict of interest.

*Acknowledgements.*    The authors would like to thank several persons who contributed to this work. Chengyu Wang (TUM) and Daniel Bar-
reiro Clemente (TUM) supported the work on the CFD simulations and polar identification. Moreover, Nady Kheirallah (TUM) contributed
to the rotor design, and Johanne Robke (TUM) assisted in the wake analysis. Last but not least, Christian Breitsamter, Florian Heckmeier
and Kyle Jones supported the wind tunnel measurements at TUM and UTD, respectively.



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

## Nomenclature

| | |
|---|---|
| $a$ | Axial induction factor |
| 760  $c$ | Chord length |
| $c_s$ | Speed of sound |





|  | $C_L$ | Lift coefficient |
|---|---|---|
|  | $C_D$ | Drag coefficient |
|  | $C_P$ | Power coefficient |
| 765 | $C_Q$ | Torque coefficient |
|  | $C_T$ | Thrust coefficient |
|  | $D$ | Rotor diameter |
|  | $I$ | Turbulence intensity |
|  | $I_{add}$ | Added turbulence intensity |
| 770 | $k$ | Kolmogorov constant |
|  | $M$ | Mach number |
|  | $R$ | Rotor radius |
|  | Re | Reynolds number |
|  | $U$ | Ambient wind speed (time averaged) |
| 775 | $u$ | Streamwise velocity component (time averaged) |
|  | $v$ | Lateral velocity component (time averaged) |
|  | $w$ | Vertical velocity component (time averaged) |
|  | $\alpha$ | Angle of attack |
|  | $\beta$ | Pitch angle |
| 780 | $\theta$ | Twist angle |
|  | $\gamma$ | Wind misalignment angle |
|  | $\epsilon_k$ | Dissipation rate |
|  | $\phi$ | Flow angle |
|  | $\rho$ | Air density |
| 785 | $\Gamma$ | Circulation |
|  | $\Omega$ | Rotor angular speed |
|  | ALM | Actuator-line method |
|  | BEM | Blade element momentum |
|  | CFD | Computational fluid dynamics |
| 790 | LES | Large-eddy simulation |
|  | S-PIV | Stereo-Particle image velocimetry |