# Peer review of "Design, steady performance and wake characterization of a scaled wind turbine with pitch, torque and yaw actuation"

_Wind Energy Science, 2021_

## Author Response (AR1)

**REVISION TO MANUSCRIPT DRAFT**

**Wind Energy Science Discussion**

**Design, performance and wake characterization of a scaled wind turbine**

**(now changed into "Design, steady performance and wake characterization of a scaled wind turbine with pitch, torque and yaw actuation")**

The authors would like to thank the two reviewers for their time and for the useful feedback. All inputs that they provided have contributed to the improvement of the paper. In addition, we have taken the opportunity of this revision to make several editorial changes in order to improve readability, and we have expanded the text at various points throughout the manuscript to improve clarity. For completeness, we have also added a description of the yaw system, which was missing in the previous version of the manuscript; this has implied the addition of a new co-author, Dr. Franz Mühle.

A revised version of the paper is attached to the present reply, with the main changes highlighted in red (deletions) and blue (additions).

A list of point-by-point replies to the reviewers' comments is reported in the following.

**Reviewer #1**

1.   **[Reviewer]** *I would appreciate if you could list a few references related to this statement "There is only a handful of articles that address the methodology behind the design of scaled models and/or provide some assessment of their characteristics".*
**[Authors]** Thank you for the comment. In reality, in the paragraphs following this sentence we make an extensive literature review on this topic, where we refer, to the best of our knowledge, to almost all works on the design and characterization of scaled wind turbine models. So we believe that the present version of the text already addresses the reviewer's comment.

2.   **[Reviewer]** *Figure 2 : Could you provide some other parameters of the XFOIL simulation, e.g. Ncrt, Ma, etc.?*
**[Authors]** This figure was removed from the revised version of the manuscript. However, we have added the information requested by the reviewer in Fig. 8 of the new paper version.

3.   **[Reviewer]** *L160: Can you provide a reference to the last statement?*
**[Authors]** We added a reference.

4.   **[Reviewer]** *L176: How are CL and W estimated or calculated to do this process?*
**[Authors]** We clarified this point in the revised version.

5.   **[Reviewer]** *L212: Until now the presentation is based on design and assembly, what "typical use" are you referring to?*
**[Authors]** Thank you for the comment, we agree that the original statement is vague. We modified the sentence to improve clarity.

6.   **[Reviewer]** *L232: Strouhal number not frequency. Please clarify what are f, D and U here.*
**[Authors].** Thank you, indeed, the Strouhal number is a non-dimensional frequency. We modified the text and clarified the meaning of the variables.

7.   **[Reviewer]** *Fig. 7: How far upstream of the turbine are these profiles measured? It would be nice if you could add the measured TI-profiles.*
**[Authors]** We agree that this is an important piece of information, which was missing in the original manuscript. We added the profiles and information about the inflow measurement position in the text.

8.   **[Reviewer]** *L243: Only G1 was provided with a reference so far. Could you include a reference to G2?*
**[Authors]** We added a reference in the introduction.

9.   **[Reviewer]** *L299: How is the inflow speed measured? The technique, sensor and details are missing.*
**[Authors]** Thank you for the comment. This part is not referring to a specific inflow measurement, but on the data acquisition capabilities. We modified the text to clarify this point.

10.   **[Reviewer]** *L302: Is there any kind of filtering in the data previous/after the controlling operations?*
**[Authors]** Thank you for the comment, we added a comment on filtering to the revised text.

11.   **[Reviewer]** *L330: The overheat ratio should be 1.8 (from the reference mentioned)*
**[Authors]** We corrected the sentence.

12.   **[Reviewer]** *L333: You state that one wind tunnel creates a „uniform velocity profile ". However, from my understanding both velocity profiles are sheared, i.e. non-uniform in Fig.7.*
**[Authors]** Figure 7 shows only TUM-HT and UTD-HT profiles. We modified the text to clarify this point.

13.   **[Reviewer]** *L333: What does "Clean tunnel" exactly mean? Spires, roughness, turbine, everything out?*
**[Authors]** We clarified this point in the revised version of the text.

14.   **[Reviewer]** *L12: "…with a similar slightly larger scaled model turbine": for the sake of better readability I would rephrase this, e.g. „…with a similar **but** slightly larger scaled model turbine".*
**[Authors]** Thank you for the comment. We implemented the suggested modification.

15.   **[Reviewer]** *Caption Fig. 19: there is a double "the the".*
**[Authors]** We corrected this typo.

16.   **[Reviewer]** *L561: I suggest replacing "represented" by „represent.*
**[Authors]** We modified the text as suggested.

17.   **[Reviewer]** *L564: remove "at" from "…including at the rotor…" and „in" from „..in the near- and far-wake regions..*
**[Authors]** We modified the text as suggested.

18.   **[Reviewer]** *L564: insert "spanwise" before "circulation"*
**[Authors]** We modified the text as suggested.

19.  **[Reviewer]** *L571: remove "an" from "have an almost"*
**[Authors]** We modified the text as suggested.

**Reviewer #2**

1.  **[Reviewer]** *The title mentions "closed-loop controls", whereas these aspects are not characterised or validated here. Only specifications of actuators are given.*
**[Authors]** We agree that the title was misleading: the model is designed with pitch, torque and yaw actuation, but the present study is limited to the characterization of the model performance in steady operating conditions. We have more clearly stated the scope of the paper by modifying the text in various places throughout the manuscript. Additionally, we have changed the title to *"Design, steady performance and wake characterization of a scaled wind turbine with pitch, torque and yaw actuation"*. An alternative title could be *"Design, steady performance and wake characterization of a scaled wind turbine"*, but we prefer the former because the presence of these three actuations has profound implications on the mechanical and electrical design of the model and of its software, as explained in various places throughout the paper.

2.  **[Reviewer]** *P5, L120-121: "In fact, wake behaviour is independent from the rotor-based Reynolds number when this parameter is larger than circa $10^5$": isn't it true only for the far-wake? Are Chamorro's results universal enough to be considered as a proof of this independency?*
**[Authors]** Thank you for the comment. We agree, and we modified the text to clarify this points.

3.  **[Reviewer]** *P5, L125 : "Considering these various requirements and constraints, the rotor diameter was finally chosen as D= 0.6 m." : this statement seems a bit straightforward since no Reynolds number value had been given so far for the present model wind turbine. Another key parameter to choose the rotor diameter is the blockage ratio. A discussion on the targeted facilities is needed here*
**[Authors]** Thank you for the remark, and we agree with it. We expanded the text and moved the discussion on the target wind tunnels here.

4.  **[Reviewer]** *P6, from L136: The dynamic similarity is finally not respected since you have an efficiency of 30 instead of 120 in full scale. That should be clearly stated at the end of this paragraph.*
**[Authors]** Thank you for the comment. We already explained in the original text that the whole rotor design philosophy is based on the fact that the dynamic similarity cannot be fulfilled. However, we now also added it explicitly at the end of the paragraph.

5.  **[Reviewer]** *P6, L147-149: "In addition, as shown in Fig. 2, at these low Reynolds a standard wind turbine airfoil as S-806 (Tangler, 1987) suffers from multiple stall-reattachment cycles even at small angles of attack.": one cannot deduce physical explanations on the flow topology by looking at these results coming from Xfoil. One can just say that Xfoil calculations do not converge at these Reynolds numbers. In Xfoil, there are boundary layer transition criteria, how were they set up? It is also important to stress that the Reynolds number will evolve a lot along the blade span and that the given values correspond generally to the maximum Reynolds number that will be encountered at tip blade.*
**[Authors]** We agree that a more thorough discussion on the Xfoil calculations would be necessary. However, we think that this would add unnecessary details and would not help the reader focus on the essence of that paragraph, which is the fact that full scale airfoils do not, typically, work well at low Reynolds number. Therefore, we removed that figure and adjusted the text accordingly.

6.    **[Reviewer]** *P8, L186 : "The power coefficient is estimated by using BEM": give more details on the BEM computations (parameters, correction options, etc).*
**[Authors]** We added a reference and expanded the description.

7.    **[Reviewer]** *Figure 7: plot also the turbulence intensity profiles in both facilities. They should not be uniform and this information is important for the discussion on the wake added turbulence intensity.*
**[Authors]** Thank you for the comment, we agree that this is necessary and we added that figure.

8.    **[Reviewer]** *P15, Table 2: the blockage ratio in TUM wind tunnel is 5.8% and not 5%. It is therefore above the threshold of 5%, which is commonly accepted as the guarantee to prevent blockage effects. Check the value and adapt the discussion to it.*
**[Authors]** Thank you for spotting this, we changed the value and adapted the discussion.

9.    **[Reviewer]** *Page 17, Fig. 9: Give more details on the chosen parameters (lambda, spanwise location, etc). The indications of velocity should be based on the relative velocity that the blade will experience, not on the inflow velocity only. Additionally, this relative velocity will probably be calculated at the blade tip thanks to the TSR and so, corresponds to the maximum relative velocity that the blade will experience. Reformulate the discussion accordingly.*
**[Authors]** We modified the text to improve clarity on this topic.

10.   **[Reviewer]** *Page 18, Fig 10: why the maximum of power coefficient is 0.3 here and 0.4 on Fig 8a? Because HT instead of LT? Would it be the same in LT? Where the blade imperfections could be more influencing?*
**[Authors]** Thank you for the comment. In the paper, Figure 10b showed measurements performed at 8 m/s and with TUM-HT inflow, which actually show a maximum $C_P$ of 0.3 instead of 0.4. We have checked once more the experimental results used for generating the plots within the figure, and we found that they were affected by a systematic error in the wind speed measurement that could not, however, be corrected a posteriori. We have modified the figure (now 9b), which now shows measurements taken at 10 m/s with TUM-HT inflow. The reported maximum $C_P$ is about 0.4, in line with the maximum $C_P$ observed with UTD-LT inflow (Fig 8a).

11.   **[Reviewer]** *P20, L413: "Figure 14 shows the downstream evolution of the velocity deficit at wake center for different thrust coefficients": how is determined the wake center? Give also the value of the wake growth parameter*
**[Authors]** Thank you for the comment. We explained the calculation of the wake center in the revised version of the text.

12.   **[Reviewer]** *P23, L465, Eqs 7: please define more precisely the turbulence intensity: streamwise turbulence intensity? Horizontal turbulence intensity? Total turbulence intensity? Additionally, the subtraction of the turbulence intensity at hub height is relevant for uniform flow conditions. However, for BL flows, the turbulence intensity is non-uniform. This will fully bias the discussion on the added turbulence intensity. I would recommend subtracting the turbulence intensity profile of the inflows.*
**[Authors]** Thank you for catching this mistake in the discussion. The calculation was indeed done by subtracting the two TI profiles and not just a point at hub height, but the equation was wrong. We have now fixed the figure, changed the text and added the definition of TI.

13.   **[Reviewer]** *P26, L506-508: "This is in agreement with previous studies (Bastankhah and Porté-Agel, 2017a) and in line with the observation that the breakdown of the tip vortices, which occurs at*

*approximately x/D = 4, removes a separation layer between the wake and the ambient flow, thereby facilitating the exchange of momentum (Medici, 2006).": The second statement is not proven here.*
**[Authors]** It is true that the second statement is not proven here. It is however a good explanation of why the turbulent momentum fluxes are stronger after x/D 3-4. We modified the text accordingly.

14. **[Reviewer]** *P30, L586: "The turbulence dissipation rate was also characterized in this work, for the first time directly from wind tunnel measurements.": it is not the first time. However, maybe the authors meant "the first time for our research group"?*
**[Authors]** Thank you for the remark. We modified the text accordingly.

15. **[Reviewer]** *P2,L39: give the blockage ratio*
**[Authors]** We added the blockage ratio.

16. **[Reviewer]** *P2, L47 : "The rotor aerodynamics is"*
**[Authors]** Thank you for the remark. We modified the text as suggested.

17. **[Reviewer]** *At several locations "Reynolds number" instead of "Reynolds*
**[Authors]** We modified the text as suggested.

18. **[Reviewer]** *P3, L78: reformulate "Against this background". Because of this background? In continuity of this background?*
**[Authors]** Thank you for the comment. We modified the text.

19. **[Reviewer]** *P8, L195: "The rated rotor speed of the scaled model, …, was primarily determined by the requirement to avoid compressible effects over the blade, as expressed by the condition …, cs being the speed of sound.": put this information earlier in the general considerations for the design.*
**[Authors]** We believe that this information is more suitable for the Rotor Design Methodology section, since it was actually determined during this stage of the model development and not during the initial sizing that is described in the "General considerations" (2.3.1) section.

20. **[Reviewer]** *P9, L 226-229: with which tool were theses aero-elastic simulations performed?*
**[Authors]** We added the name of the tool and corresponding reference.

21. **[Reviewer]** *P12, L270 : "in the wind tunnel"*
**[Authors]** We modified the text as suggested.

22. **[Reviewer]** *P16, L349 : give the value of this optimum pitch angle, here and in the caption of Fig. 8*
**[Authors]** We added the pitch angle value.

23. **[Reviewer]** *Page 17, Fig. 9 caption: "Aerodynamic efficiency"*
**[Authors]** We modified the text as suggested.

24. **[Reviewer]** *Page 18, L384: "Figure 11a shows the airfoil efficiency as a function of angle of attack for the nominal and tuned polars." They are called "design" and "identified" polars in the figure 11 caption. Harmonize the names*
**[Authors]** Thank you for the comment. As suggested, we modified the text to harmonize the names.

25. **[Reviewer]** *P18, L384-385: "Results show that, although not identical, the difference between the two sets of polars is small, which seems to indicate a good overall manufacturing precision of the blades": Doesn't it rather indicate that the experimental model behaves as the BEM expected?*
**[Authors]** Thank you for the comment. We modified the text to clarify this point.

26. **[Reviewer]** *P19, Fig. 12: put the TUM results on the first plane, symbols are hidden by UTD results. Remove the continuous lines*
**[Authors]** We improved the figure as suggested.

27. **[Reviewer]** *P20, L410: "The profiles of the two different inflow conditions are similar": The magnitudes of the velocity deficit are similar, but not the profiles.*
**[Authors]** Thank you for the comment. We modified the text to clarify this point.

28. **[Reviewer]** *P23, Fig. 16: Change the color scale to have a better color contrast in the velocity map. No reason here to extend the color scale from -50 to +50%.*
**[Authors]** Thank you for the comment. We changed the color scale to -20 +20%.

29. **[Reviewer]** *P23, Fig. 16 caption: "in the wakes of the G06 and of the DTU 10 MW".*
**[Authors]** Thank you, we corrected that typo.

30. **[Reviewer]** *P24, L489: how is the axial induction factor determined here?*
**[Authors]** We added an explanation of the calculation of the axial induction in the revised text.

31. **[Reviewer]** *P26, Eq 9: remove the bar on density by precising that you make the hypothesis of incompressible flows.*
**[Authors]** We did not eliminate the bar, but we adjusted the text to be more precise on this point.

32. **[Reviewer]** *P29, Fig 23: plots are inverted (vertical profiles on the left and horizontal ones on the right)?*
**[Authors]** Thank you for catching this mistake, which has now been corrected in the revised version of the manuscript.

The Authors